# The role of *APETALA1* in petal number robustness

**Marie Monniaux[1†‡], Bjorn Pieper[1†], Sarah M McKim[2§],
Anne-Lise Routier-Kierzkowska[1#], Daniel Kierzkowski[1#], Richard S Smith[1],
Angela Hay[1]***

[1]Max Planck Institute for Plant Breeding Research, Cologne, Germany; [2]Plant Sciences Department, University of Oxford, Oxford, United Kingdom

**\*For correspondence:**
hay@mpipz.mpg.de

[†]These authors contributed equally to this work

**Present address:** [‡]Laboratoire Reproduction et Développement des Plantes, Université de Lyon, Lyon, France; [§]Division of Plant Sciences, School of Life Science, University of Dundee at the James Hutton Institute, Scotland, United Kingdom; [#]Institut de Recherche en Biologie Végétale, Département de Sciences Biologiques, Université de Montréal, Montreal, Canada

**Competing interests:** The authors declare that no competing interests exist.

**Abstract** Invariant floral forms are important for reproductive success and robust to natural perturbations. Petal number, for example, is invariant in *Arabidopsis thaliana* flowers. However, petal number varies in the closely related species *Cardamine hirsuta*, and the genetic basis for this difference between species is unknown. Here we show that divergence in the pleiotropic floral regulator *APETALA1* (*AP1*) can account for the species-specific difference in petal number robustness. This large effect of *AP1* is explained by epistatic interactions: *A. thaliana AP1* confers robustness by masking the phenotypic expression of quantitative trait loci controlling petal number in *C. hirsuta*. We show that *C. hirsuta AP1* fails to complement this function of *A. thaliana AP1*, conferring variable petal number, and that upstream regulatory regions of *AP1* contribute to this divergence. Moreover, variable petal number is maintained in *C. hirsuta* despite sufficient standing genetic variation in natural accessions to produce plants with four-petalled flowers.
DOI: https://doi.org/10.7554/eLife.39399.001

## Introduction

Determining the genetic basis of developmental traits, and how these evolve to generate novelties, is a major goal of evolutionary developmental biology. Developmental systems are remarkably robust to natural perturbations, such that individuals tend to develop normally despite variation in the environment or their genetic make-up (*Wagner, 2005*). Therefore, a particular challenge is to understand the developmental transitions between the robust morphology of individuals within a species, and the variation in form between species.

Petal number is a robust trait in flowering plants and usually invariant within species and even higher taxonomic orders. For example, three petals are commonly found in monocots while five petals are characteristic of many core eudicots (*Endress, 2011*; *Specht and Bartlett, 2009*). On the other hand, petal number is much more labile in basal angiosperms (*Endress, 2001*; *Endress, 2011*; *Specht and Bartlett, 2009*), suggesting that this trait was canalized during angiosperm evolution to produce a stable phenotype in the face of genetic and environmental perturbation. Petals are usually required to open the flower and to help attract pollinators (*Fenster et al., 2004*; *van Doorn and Van Meeteren, 2003*), therefore a stable number of petals could ensure a reliable display of the reproductive organs and a reproducible cue for pollinators.

The model plant *Arabidopsis thaliana* belongs to the Brassicaceae family, which are commonly called crucifers after the cross-shaped arrangement of four petals in their flowers (*Endress, 1992*). Petals acquire their identity via the combined activity of A- and B-class floral organ identity genes, according to the ABC model of flower development (*Coen and Meyerowitz, 1991*). However, less is known about the genetic control of petal number. In *A. thaliana* flowers, initiation of four petals depends on the size of the floral meristem, the establishment of boundaries that demarcate the position of petal primordia on the floral meristem, the transient formation of auxin activity maxima in

**eLife digest** Many plants produce flowers that attract insects to land on them. Different insects are attracted to flowers of different shapes and colors. Therefore, it is generally advantageous for plants of the same species to produce flowers that look very similar.

For example, a small weed known as Arabidopsis – which is often used in research studies – produces little white flowers that all have four petals. Thus, the number of petals in Arabidopsis flowers is said to be a 'robust' trait. However, a closely-related plant called hairy bittercress produces flowers with any number of petals between zero and four. Studying the genetic differences between Arabidopsis and hairy bittercress can help to reveal why the numbers of petals on hairy bittercress flowers vary.

A gene called *APETALA1* helps to control how petals form. Monniaux, Pieper et al. found that Arabidopsis and hairy bittercress have different versions of this gene that determine whether the number of petals may vary between individual flowers. Inserting the Arabidopsis version of *APETALA1* into hairy bittercress plants caused the plants to produce flowers that had more similar numbers of petals to each other, that is, the petal number became more robust.

Monniaux, Pieper et al. then used a statistical method called quantitative trait locus analysis to identify the precise location of regions in the hairy bittercress genome that control petal number. This showed that the Arabidopsis version of *APETALA1*, but not the hairy bittercress version, conceals the action of these genes that could alter petal number.

These findings reveal that evolutionary change in a single gene of hairy bittercress unmasked the action of other genes that caused petal number to vary. A next step will be to identify some of these genes and understand how they control petal number.

DOI: https://doi.org/10.7554/eLife.39399.002

these positions, and the general mechanisms of lateral organ outgrowth (*Huang and Irish, 2015*; *Huang and Irish, 2016* ; *Irish, 2008*). Consistent with this complexity, few *A. thaliana* mutants specifically affect petal number; among them is *petal loss* (*ptl*), which displays a variable loss of petals caused by mutation of the *PTL* trihelix transcription factor (*Brewer et al., 2004*; *Griffith et al., 1999*). Other mutants pleiotropically affect petal number, such as mutations in the MADS-box transcription factor *APETALA1* (*AP1*), which shows floral meristem identity defects in addition to variable petal loss (*Irish and Sussex, 1990*; *Mandel et al., 1992*).

*Cardamine hirsuta* is a close relative of *A. thaliana* that lacks a robust phenotype of four petals (*Hay et al., 2014*). Instead, *C. hirsuta* flowers display a variable number of petals, between zero and four, on a single plant (*Monniaux et al., 2016*; *Pieper et al., 2016*). This phenotype varies in response to both environmental and genetic perturbation. *C. hirsuta* flowers show seasonal variation in petal number, with spring-flowering plants producing more petals than summer-flowering plants (*McKim et al., 2017*). Seasonal cues, such as day length, winter cold, and particularly ambient temperature, all influence the number of petals produced in *C. hirsuta* flowers (*McKim et al., 2017*). Furthermore, petal number is strongly influenced by natural genetic variation in *C. hirsuta*. A polygenic architecture of small to moderate effect quantitative trait loci (QTL), that shift the trait in both directions, contribute to petal number variation in *C. hirsuta* (*Pieper et al., 2016*). Alleles of large effect and low pleiotropy were identified in genetic screens for four-petalled mutants in *C. hirsuta*, but were not detected by QTL analysis in natural accessions (*Pieper et al., 2016*). Thus, the distribution of allelic effects found in natural populations of *C. hirsuta* is more likely to maintain variation, rather than robustness, in petal number.

Petal number varies both within and between species, evolving from a robust state of four petals, typified by *A. thaliana*, to a variable state in *C. hirsuta*. Phenotypic divergence between species is necessarily derived from variation within species, but identifying these evolutionary transitions is not a straight-forward task. This is because similar phenotypes that vary within and between species may or may not be caused by similar genetic mechanisms. For example, the same light-pigmentation alleles that are fixed in a yellow-bodied *Drosophila* species, segregate in a closely related brown-bodied species and contribute to clinal variation in its body colour (*Wittkopp et al., 2009*). However, in another example, genes responsible for leaf shape differences between *A. thaliana* and *C.*

*hirsuta* were not detected as leaf shape QTL in *C. hirsuta* (**Cartolano et al., 2015**). Therefore, to understand how petal number variation is produced and how it evolved, it is important to investigate both the genetic basis of variation within species and divergence between species. For example, to address questions such as: How many genes contribute to trait divergence between species? How large are their effects? Do they have pleiotropic functions? How do they interact with genes causing variation in natural populations?

A simple prediction about robust phenotypes, such as petal number in *A. thaliana*, is that they are invariant because genetic variation is reduced by stabilizing selection on the phenotype. On the other hand, a developmental pathway might be robust because certain alleles prevent the pheno-typic effects of new mutations. This would effectively buffer the phenotype and hide underlying genetic variation. Previous studies of vulva development in *Caenorhabditis* (**Félix, 2007**), and eye development in cavefish (**Rohner et al., 2013**), support the latter view, showing that there is exten-sive, selectable genetic variation affecting robust phenotypes, which can be exposed by genetic or environmental perturbation. Moreover, studies that use gene expression as a trait, have mapped QTL that influence variance rather than mean expression level (**Hulse and Cai, 2013**), and identified selection acting on expression noise rather than mean level (**Metzger et al., 2015**). However, there are few examples (**Rohner et al., 2013**) where the genetic basis of morphological differences between species can be traced to the release of cryptic variation.

In this study, we investigate the evolutionary transition from a robust phenotype of four petals, typified by *A. thaliana*, to a variable petal number in *C. hirsuta*. We show that divergence in the pleiotropic floral regulator *AP1* can account for the difference in petal number robustness between species. This large effect of *AP1* is explained by epistatic interactions: *A. thaliana AP1* masks the phenotypic expression of all petal number QTL in *C. hirsuta* and, in this way, confers robustness. We show that *C. hirsuta AP1* fails to complement this function of *A. thaliana AP1*, conferring variable petal number, and that upstream regulatory regions of *AP1* contribute to this divergence.

## Results

### Petal number variation in *C. hirsuta* flowers

The flowers of *A. thaliana* and other Brassicaceae species are readily distinguished by their four pet-als. This phenotype is robust to natural genetic variation; for example, flowers from genetically diverse *A. thaliana* accessions consistently produce four petals (**Figure 1a**). *C. hirsuta* lacks this robustness and shows variation in petal number. For example, we found similar frequencies of each petal number between zero and four in flowers from 39 *C. hirsuta* accessions sampled from across the species range (**Figure 1b**) (**Hay et al., 2014**). Therefore, petal number varies within *C. hirsuta* and is a species-level trait that distinguishes *C. hirsuta* from *A. thaliana*.

Petals initiate in the second whorl of *A. thaliana* and *C. hirsuta* flowers, each positioned between two outer sepals, with the inner whorls being occupied by male and female reproductive organs (sta-mens and carpels) (**McKim et al., 2017**; **Smyth et al., 1990**). Small petal primordia are readily observed in the second whorl of *A. thaliana* floral buds, located between first-whorl sepals and third-whorl stamens (**Figure 1c,d**). In contrast to this, petal primordia were often missing in *C. hirsuta* flowers at similar developmental stages (**Figure 1e**). Instead, we observed a flat surface in the sec-ond whorl with no indication of outgrowths (**Figure 1f**). However, when we found petal primordia in *C. hirsuta* flowers, their development appeared indistinguishable from those in *A. thaliana* (**Fig-ure 1—figure supplement 1**). These petals occupied any of the four positions available in the sec-ond whorl, with a slightly higher frequency in abaxial positions (**Figure 1—figure supplement 2**). Therefore, the number of petals in a *C. hirsuta* flower is determined at early stages of petal initiation and outgrowth.

### Auxin activity maxima fail to form in whorl two of *C. hirsuta* floral meristems

To study the earliest stages of petal initiation, we tracked floral meristem development using time-lapse confocal laser scanning microscopy and analysed growth in these 4-dimensional image stacks (**Barbier de Reuille et al., 2015**). We followed the formation of auxin activity maxima during petal initiation in *A. thaliana* and *C. hirsuta* using the *DR5::VENUS* and *DR5v2::VENUS* auxin activity

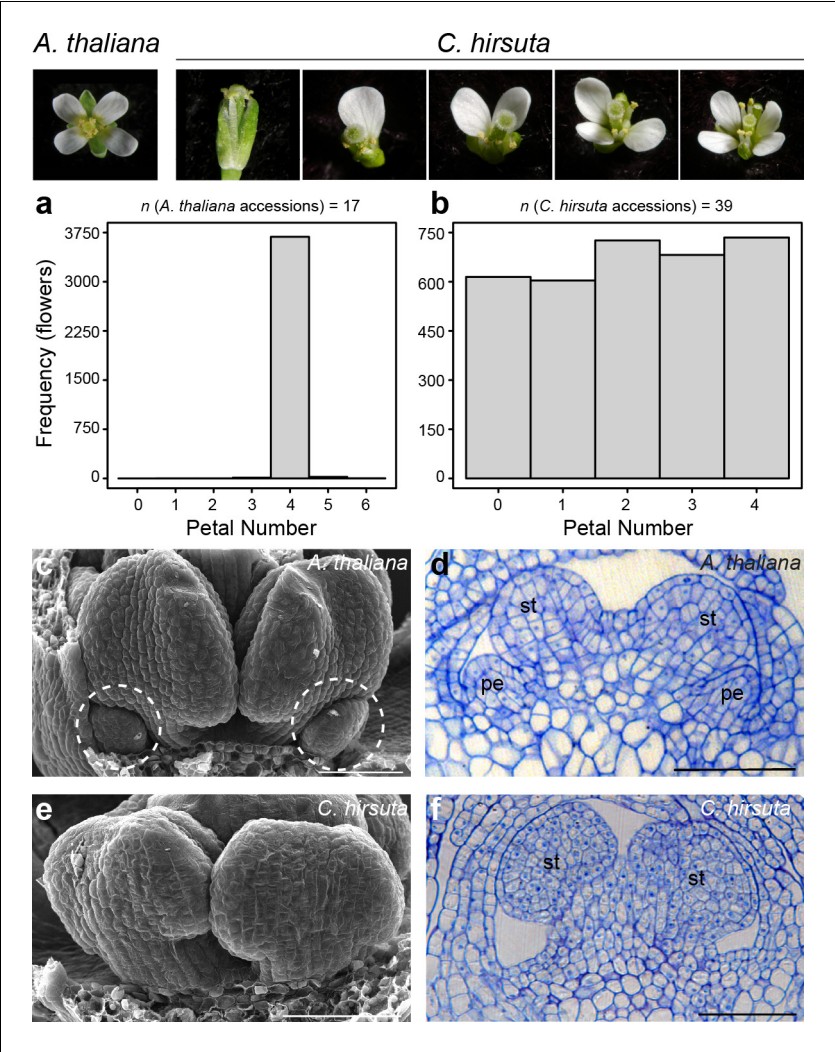

**Figure 1.** Petal number is robust in *A. thaliana* and variable in *C. hirsuta*. Four-petalled *A. thaliana* flower compared with *C. hirsuta* flowers containing 0, 1, 2, 3 or 4 petals. (a, b) Histograms showing petal number on the x-axis and frequency of flowers of the y-axis for (a) 17 *A. thaliana* accessions (n = 3725 flowers from 149 plants) and (b) 39 *C. hirsuta* accessions (n = 3362 flowers from 143 plants). (c, e) Scanning electron micrographs of stage eight flowers with covering sepals dissected away to show medial stamen primordia and small petal primordia (dashed circles) present in *A. thaliana* (c) and absent in *C. hirsuta* (e). (d, f) Longitudinal sections of stage eight flowers showing small petal primordia present in *A. thaliana* (d) and absent in *C. hirsuta* (f). Abbreviations: pe, petal; st, stamen. Scale bars: 20 μm (c–f).

DOI: https://doi.org/10.7554/eLife.39399.003

The following figure supplements are available for figure 1:

**Figure supplement 1.** Scanning electron micrograph of a *C. hirsuta* stage eight flower.

DOI: https://doi.org/10.7554/eLife.39399.004

**Figure supplement 2.** Petal position in *C. hirsuta* flowers (n = 144 flowers, flower numbers 1 – 25).

DOI: https://doi.org/10.7554/eLife.39399.005

sensors (*Barkoulas et al., 2008*; *Heisler et al., 2005*; *Liao et al., 2015*). At sites of petal initiation in *A. thaliana*, auxin activity maxima formed in 2 – 3 epidermal cells on the floral meristem flank prior to growth of these cells (*Figure 2a,b*; 83% *DR5::VENUS* observation rate, *Figure 2—figure supplement 1*). However, in *C. hirsuta*, auxin activity maxima often failed to form on the floral meristem, and instead were either absent or aberrantly located in the first whorl between sepals (*Figure 2c,d*; 36% *DR5v2::VENUS* observation rate, *Figure 2—figure supplement 1*). Therefore, four sites of petal

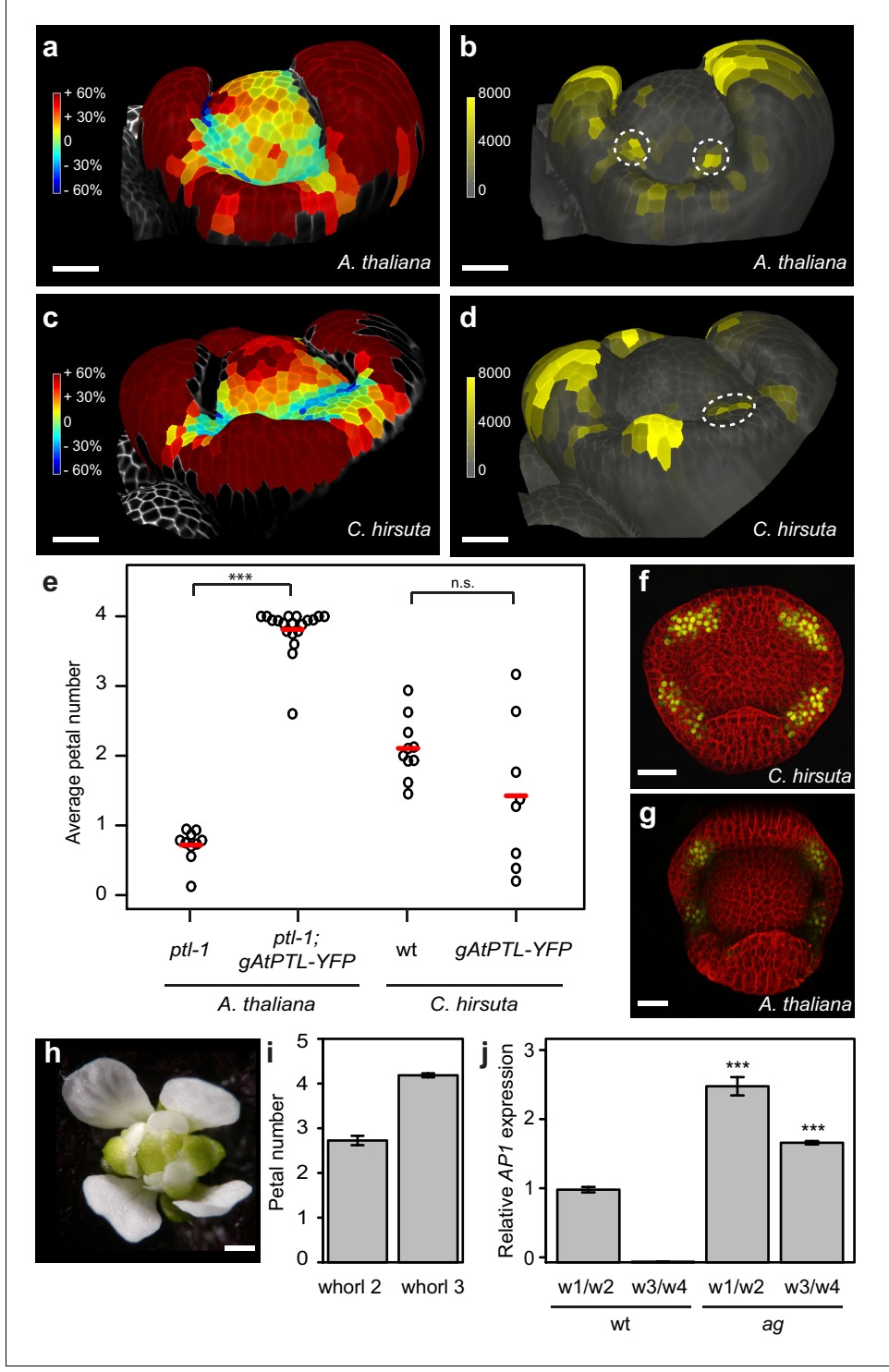

**Figure 2.** Auxin activity maxima fail to form in whorl two of *C. hirsuta* floral meristems. (**a–d**) Heat maps of change in relative cell area of floral primordia over 24 hr of growth (**a, c**) and surface projections of *DR5::VENUS* expression (**b, d**) in *A. thaliana* (**a, b**) and *C. hirsuta* (**c, d**). Colour bars: percentage increase (warm colours) and decrease (cool colours) of cell area (**a, c**) and signal intensity (yellow) in arbitrary units (**b, d**). Dashed circles indicate expression maxima that correspond to initiating petals. Floral primordia are shown in side view facing a lateral sepal. (**e**) Beeswarm plot of average petal number in *A. thaliana ptl-1* (*n* = 149 flowers, 10 plants) and *ptl-1; AtPTL::AtPTL:YFP* (*n* = 266 flowers, 19 independent insertion lines), and *C. hirsuta* Ox (*n* = 145 flowers, 10 plants) and *AtPTL::AtPTL:YFP* (*n* = 110 flowers, eight independent insertion lines). Red lines indicate means. (**f, g**) CLSM
*Figure 2 continued on next page*

*Figure 2 continued*

projections showing *AtPTL::AtPTL:YFP* expression (yellow) in the regions between sepals in stage 3–4 flowers of *C. hirsuta* (f) and *A. thaliana* (g). (h) *C. hirsuta ag* flower. (i) Barplot of mean petal number in whorls 2 and 3 of *C. hirsuta ag* flowers (n = 136 flowers, four plants). Note that mean stamen number is distributed between 4 – 5 in *C. hirsuta*, reflecting variation in lateral stamen number (*Hay et al., 2014*), and third whorl petals show similar variation in *ag*. (j) Relative expression of *C. hirsuta AP1* in floral organs pooled from whorls 1, 2 (w1/w2) and whorls 3, 4 (w3/w4), in Ox compared to *ag* flowers, determined by quantitative RT-PCR and reported as means of three biological replicates (Student's *t*-test: p<0.001). Error bars represent s.e.m. Scale bars: 20 μm (a–d, f–g), 0.5 mm (h).

DOI: https://doi.org/10.7554/eLife.39399.006

The following figure supplements are available for figure 2:

**Figure supplement 1.** CLSM time-lapse series of *DR5::VENUS* in *A. thaliana* and *DR5v2::VENUS* in *C. hirsuta* flowers.
DOI: https://doi.org/10.7554/eLife.39399.007
**Figure supplement 2.** Representative flowers.
DOI: https://doi.org/10.7554/eLife.39399.008
**Figure supplement 3.** In situ hybridization of *C. hirsuta PTL*.
DOI: https://doi.org/10.7554/eLife.39399.009

initiation are usually marked by auxin activity maxima in the second whorl of the floral meristem in *A. thaliana*, but not *C. hirsuta*.

The *ptl* mutant in *A. thaliana* mimics the variable petal number found in *C. hirsuta* and shows a similar distribution of auxin activity during petal initiation as wild-type *C. hirsuta* (*Figure 2d*) (*Griffith et al., 1999*; *Lampugnani et al., 2013*). Therefore, we tested whether differences in *PTL* function could explain why petal number is robust in *A. thaliana* but variable in *C. hirsuta*. A functional fusion protein of *A. thaliana* PTL (AtPTL::AtPTL:YFP) was sufficient to restore four petals in the *ptl* mutant, but did not alter petal number when expressed in *C. hirsuta* (*Figure 2e*; *Figure 2—figure supplement 2*). Given that AtPTL::AtPTL:YFP expressed correctly in the regions between sepals in *C. hirsuta* and *A. thaliana* flowers (*Figure 2f,g*), similar to the endogenous *PTL* transcripts in *C. hirsuta* and *A. thaliana* (*Figure 2—figure supplement 3*) (*Brewer et al., 2004*; *Lampugnani et al., 2012*; *Lampugnani et al., 2013*), these results indicate that differences in *PTL* function are unlikely to account for the variation in petal number between *C. hirsuta* and *A. thaliana*.

Petals are defined by both their identity and position within a flower. To test whether the variable number of petals in *C. hirsuta* is dependent on their identity or on the location where they arise in the second whorl, we used the homeotic mutant *agamous* (*ag*), to alter floral organ identity. In *C. hirsuta ag* mutants, four petals replaced the four stamens normally found in the third whorl of wild-type flowers, while petal number remained variable and lower than four in the second whorl (*Figure 2h,i*). This means that floral organs with petal identity show no variation in number if they arise outside the second whorl. As predicted by the ABC model (*Coen and Meyerowitz, 1991*), we found *AP1* ectopically expressed in third whorl petals of *ag* mutants in *C. hirsuta* (*Figure 2j*). Therefore, ectopic expression of *AP1* is associated with an invariant number of petals, whereas endogenous *AP1* expression in whorl two of the *C. hirsuta* floral meristem is associated with variable petal number.

## *A. thaliana AP1* confers robust petal number in *C. hirsuta* and masks natural variation

We reasoned that *AP1* might be a good candidate to contribute to petal number variation in *C. hirsuta*, particularly given that *ap1* mutants in both *C. hirsuta* and *A. thaliana* show variable petal loss (*Bowman et al., 1993*; *Monniaux et al., 2017*). To test this possibility, we used a genomic construct of *A. thaliana AP1* (*AtAP1::AtAP1:GFP* (*Urbanus et al., 2009*)), which was sufficient to restore four petals in the *ap1* mutant (Mann-Whitney *U* test, p=1.07e-06) and eliminate the ectopic flowers that characterize the partial loss of floral meristem identity of *ap1* mutants (Mann-Whitney *U* test, p=2.92e-06; *Figure 3a–c*). We found that this transgene was sufficient to convert *C. hirsuta* petal number from variable to robust, elevating petal number towards the *A. thaliana* value of four petals (pairwise Mann-Whitney *U* test with Bonferroni correction, p=2.4e-08; *Figure 3a,d,e*). In contrast to

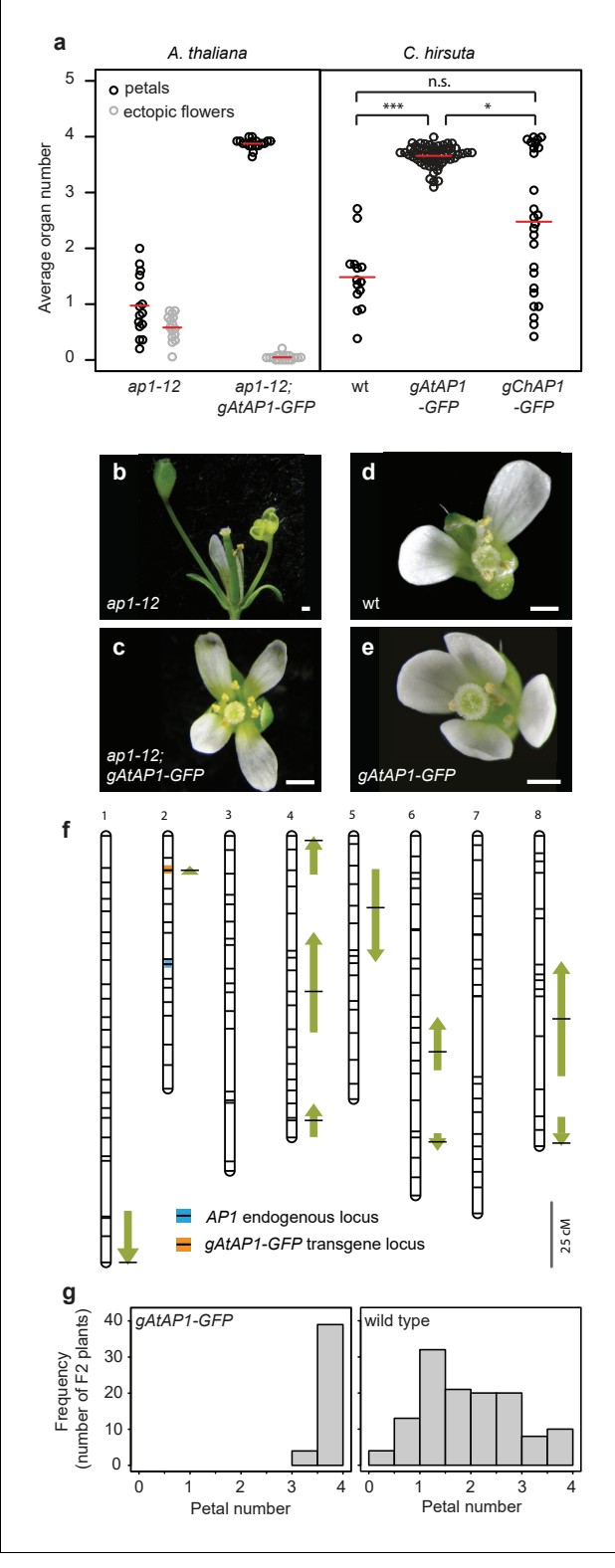

**Figure 3.** *A. thaliana AP1* confers robust petal number in *C. hirsuta* and masks natural variation. (**a**) Beeswarm plot of average petal number (black) and average number of ectopic flowers (grey) in *A. thaliana ap1-12* (n = 375 flowers, 15 plants) and *ap1-12; gAtAP1-GFP* (*AtAP1::AtAP1:GFP*; n = 472 flowers, 19 plants, two independent insertion lines), and average petal number (black) in *C. hirsuta* wild-type (wt; n = 331 flowers, 14 plants), *gAtAP1-GFP* (n = 1286 flowers, 57 plants, five independent insertion lines) and *gChAP1-GFP* (*ChAP1::ChAP1:GFP)* (n = 628

*Figure 3 continued on next page*

*Figure 3 continued*

flowers, 26 plants, two independent insertion lines). Red lines indicate means. Differences between *C. hirsuta* genotypes assessed by pairwise Mann-Whitney U test with Bonferroni correction: \*\*\*p=2.4e-08; \*p=0.015; n.s. p=0.07. (**b–e**) Representative flowers of *A. thaliana ap1-12* (**b**), *ap1-12; gAtAP1-GFP* (**c**), and *C. hirsuta* wild type (**d**), *gAtAP1-GFP* (**e**). Scale bars: 0.5 mm. (**f**) QTL for average petal number detected in the *C. hirsuta* Ox *gAtAP1-GFP* × Nz F2 mapping population are shown as arrows on the 8 chromosomes of the genetic map. Positions with the most significant effects are indicated by horizontal black lines and the length of the arrows is scaled to the 2 (Log(p)) interval for each QTL. Arrow direction indicates whether the Ox allele for each QTL increases (upward pointing) or decreases (downward pointing) petal number. Positions of the *AP1* endogenous locus (blue line) and the *gAtAP1-GFP* transgene (orange line) are indicated on the genetic map. Scale bar: 25 cM. (**g**) Distribution of average petal number in plants of the Ox × Nz F2 population that segregate homozygous for the *gAtAP1-GFP* transgene (left histogram) or without the transgene (right histogram).

DOI: https://doi.org/10.7554/eLife.39399.010

this, the distribution of petal number remained variable in *C. hirsuta* lines expressing the endogenous *AP1* locus (*ChAP1::ChAP1:GFP*; pairwise Mann-Whitney *U* test with Bonferroni correction, p=0.07: *Figure 3a*). This suggests that the function of the endogenous *AP1* locus to confer four petals is attenuated in *C. hirsuta*. Therefore, divergence in *AP1* function likely contributed to the variation in petal number between *A. thaliana* and *C. hirsuta*.

This raises the question whether petal number variation both between and within species may be caused by similar genetic changes. If *AP1* divergence contributed to petal number variation between species, do *AP1* polymorphisms contribute to this phenotype within *C. hirsuta*? To address this question, we inspected the locations of petal number QTL previously identified in five mapping populations derived from bi-parental crosses of different *C. hirsuta* accessions (*Pieper et al., 2016*), and an additional population constructed in this study (*Figure 3f*). We found that none of the QTL mapped to the *AP1* locus, which was represented by a specific genetic marker on chromosome 2 (*Figure 3f*, *Table 1*) (*Pieper et al., 2016*). Therefore, allelic variation at *AP1* does not contribute to the quantitative variation in petal number mapped in *C. hirsuta*.

However, an alternative possibility is that *AP1* divergence indirectly caused petal number to vary within *C. hirsuta* by altering the robustness of this phenotype to genetic variation. Given that petal number is a canalized trait in *A. thaliana* and robust to genetic variation (*Figure 1a*), we hypothesized that *AP1* divergence may have decanalized petal number in *C. hirsuta*, giving phenotypic expression to formerly cryptic variation (*Figure 1b*) (*Félix, 2007*; *Paaby and Rockman, 2014*). A key prediction of this hypothesis is that *A. thaliana AP1* should canalize petal number in *C. hirsuta* via

**Table 1.** *A. thaliana AP1* masks the effects of *C. hirsuta* petal number QTL

| QTL | Chromosome | Position | QTL effects | | gAtAP1 homozygous plants | |
| | | (cM) | Wild-type plants | | | |
| | | | additive | dominance | additive | dominance |
| --- | --- | --- | --- | --- | --- | --- |
| Q1 | 1 | 147.8 | −0.18 | 0.28 | - | - |
| Q2 (*gAtAP1*) | 2 | 11.9 | n.a. | n.a. | n.a. | n.a. |
| Q3 | 4 | 1.6 | - | 0.32 | - | - |
| Q4 | 4 | 53.9 | −0.19 | - | - | - |
| Q5 | 4 | 98.6 | −0.33 | - | - | - |
| Q6 | 5 | 25.7 | 0.28 | - | - | - |
| Q7 | 6 | 74.8 | −0.28 | - | - | - |
| Q8 | 6 | 116.6 | 0.82 | - | - | - |
| Q9 | 8 | 63.5 | −0.27 | - | - | - |
| Q10 | 8 | 105.6 | 0.38 | - | - | - |

n.a. – not available because the effects of all other QTL were determined conditional on zygosity at this QTL. '– '– No significant effect.

DOI: https://doi.org/10.7554/eLife.39399.011

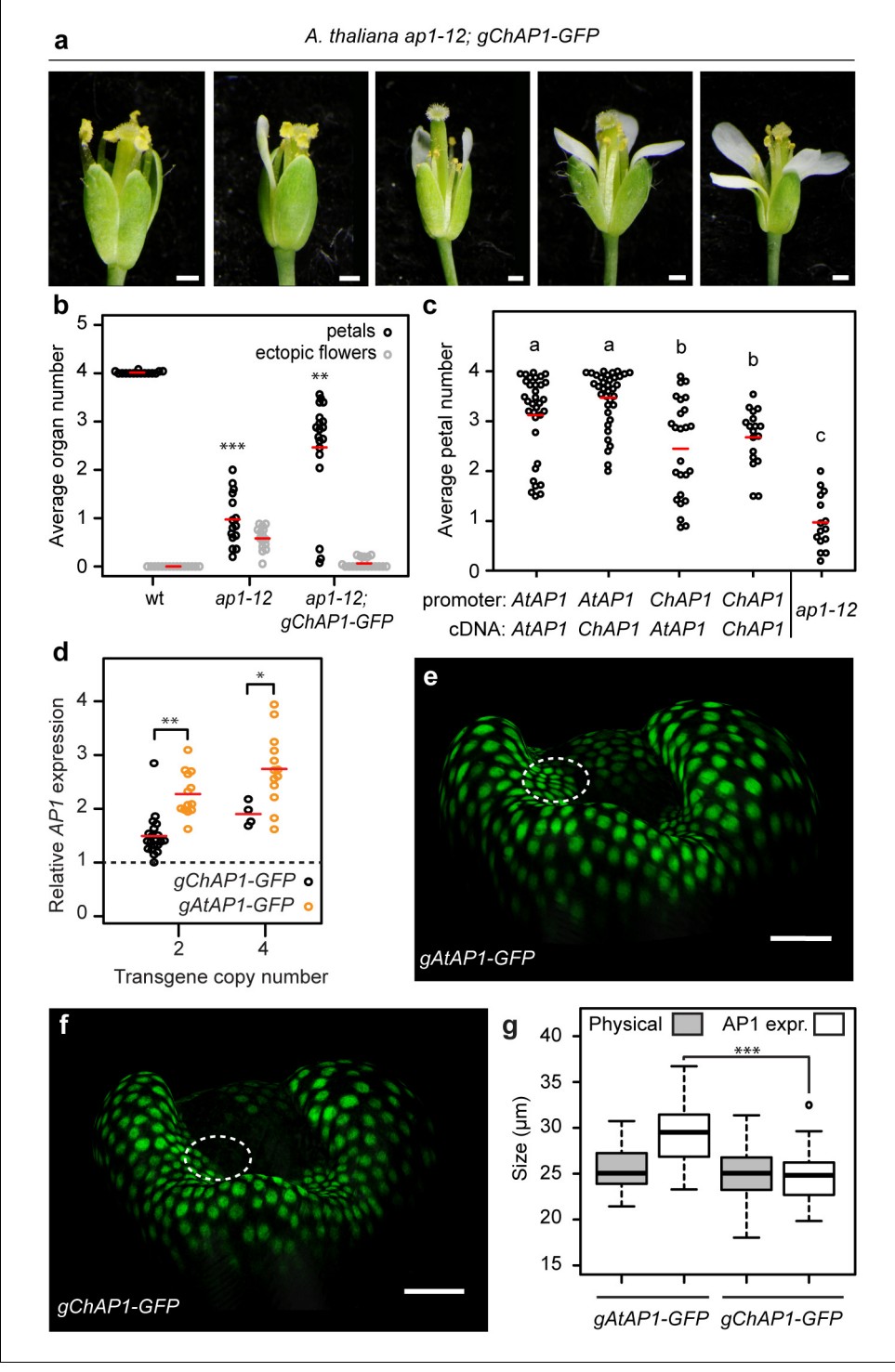

**Figure 4.** *AP1* divergence contributes to species-specific petal number. (a) *A. thaliana* flowers of *ap1-12; gChAP1-GFP* (*ChAP1::ChAP1:GFP*) genotype. (b) Beeswarm plot of average petal number (black) and average number of ectopic flowers (grey) in *A. thaliana* wild-type (*n* = 369 flowers, 15 plants), *ap1-12* (*n* = 375 flowers, 15 plants) and *ap1-12; gChAP1-GFP* (*n* = 598 flowers, 25 plants, two independent insertion lines). Red lines indicate means. Levene's test showed that the variance in petal number differed between wild-type and *ap1-12* (***p=5.588e-05) and wild-type and *ap1-12; gChAP1-GFP* (**p=0.005823), but not between *ap1-12* and *ap1-—12; gChAP1-GFP* (p=0.282). (c) Beeswarm plot of average petal number in *A. thaliana ap1-12* plants either untransformed or transformed with chimeric *AP1* constructs comprising the promoter sequences from either *AtAP1* (*A. thaliana*) or *ChAP1* (*C. hirsuta*) and the cDNA sequences from either *AtAP1* or *ChAP1*. A two-way ANOVA test on ranked data
*Figure 4 continued on next page*

*Figure 4 continued*

showed a significant effect of promoter sequence on petal number (p=9.45e-08) but no effect of coding sequence (p=0.103) and no interaction effect between the promoter and coding sequences (p=0.258). Post-hoc Tukey's HSD tests showed that *AP1* constructs containing the *A. thaliana* promoter had significantly higher petal number than those containing the *C. hirsuta* promoter, and all transgenic genotypes had higher petal number than *ap1-12* at 0.05 level of significance. *ap1-12: n* = 375 flowers, 15 plants; *ap1-12; pAtAP1::AtAP1: n* = 1454 flowers, 37 plants, 10 independent insertion lines; *ap1-12; pAtAP1::ChAP1: n* = 1414 flowers, 36 plants, nine independent insertion lines; *ap1-12; pChAP1::AtAP1: n* = 986 flowers, 25 plants, five independent insertion lines; *ap1-12; pChAP1:: ChAP1: n* = 717 flowers, 18 plants, five independent insertion lines. (**d**) Beeswarm plot of relative *AP1* expression levels in inflorescences of *C. hirsuta* transgenic lines of *gChAP1-GFP* (black) and *gAtAP1-GFP* (orange) with 2 or four transgene copies. *AP1* expression is quantified by qRT-PCR in three biological replicates of each sample and expressed relative to *AP1* expression in wild-type inflorescences (dashed line). Relative *AP1* expression is higher for *gAtAP1-GFP* lines, both for two (Student's *t*-test, p<0.01) and four (p<0.05) transgene copies. *n* = 26 plants from six independent insertion lines for *gAtAP1-GFP*; n = 24 plants from five independent insertion lines for *gChAP1-GFP*. (**e–f**) Surface projections showing nuclear expression (green) of *gAtAP1-GFP* (**d**) and *gChAP1-GFP* (**e**) in stage 4 *C. hirsuta* flowers viewed from the lateral sepal. The dashed circle indicates the petal initiation domain on the floral meristem. (**g**) Boxplot of the size of inter-sepal regions (Physical) and the extent of AP1 expression along these transects into whorl 2 (AP1 expr) in *C. hirsuta* stage four floral meristems of *gAtAP1-GFP* and *gChAP1-GFP*. Size of the AP1 expression domain differs significantly between genotypes (Wilcoxon test, p<0.001; *n* = 7 samples each genotype) but physical size does not (p=0.44). Box and whiskers: quartiles, circles: outliers, black lines: median. Scale bars: 0.5 mm (**a**), 20 μm (**e, f**).
DOI: https://doi.org/10.7554/eLife.39399.012

The following figure supplements are available for figure 4:

**Figure supplement 1.** Quantification of ectopic flowers produced by chimeric *AP1* constructs complementing *A. thaliana ap1-12*.
DOI: https://doi.org/10.7554/eLife.39399.013

**Figure supplement 2.** Example of a time-lapse series for *AtAP1::AtAP1:GFP* and *ChAP1::ChAP1:GFP* flowers in *C. hirsuta*.
DOI: https://doi.org/10.7554/eLife.39399.014

**Figure supplement 3.** Quantitative image analysis of *C. hirsuta* stage-4 flower meristems expressing *AtAP1:: AtAP1:GFP*.
DOI: https://doi.org/10.7554/eLife.39399.015

**Figure supplement 4.** Measurements of the physical boundary size and the AP1 expression boundary size (as defined in *Figure 4—figure supplement 3*) in 7 samples of *AtAP1::AtAP1:GFP* and *ChAP1::ChAP1:GFP* stage-4 flowers in *C. hirsuta*.
DOI: https://doi.org/10.7554/eLife.39399.016

**Figure supplement 5.** Variability of measurements of the physical boundary size and the AP1 expression boundary size, as defined in *Figure 4—figure supplement 3*.
DOI: https://doi.org/10.7554/eLife.39399.017

**Figure supplement 6.** Measurements of the lateral and median lengths of the flower and of the meristem, and the meristem area (as defined in *Figure 4—figure supplement 3*) in 7 samples of *AtAP1::AtAP1:GFP* and *ChAP1:: ChAP1:GFP* stage-4 flowers in *C. hirsuta*.
DOI: https://doi.org/10.7554/eLife.39399.018

**Figure supplement 7.** CLSM projections of *C. hirsuta* (**a**) and *A. thaliana* (**b**) stage four flowers co-expressing *gAtAP1-GFP* (green) and *gChAP1-RFP* (red).
DOI: https://doi.org/10.7554/eLife.39399.019

epistatic interactions with petal number QTL. We tested this genetic prediction in an F2 population created by crossing *C. hirsuta* Ox containing the *A. thaliana AP1* genomic locus (*AtAP1::AtAP1: GFP*), with the Nz accession (*Figure 3f*). We detected nine petal number QTL in addition to the *A. thaliana AP1* transgene locus (*Table 1*, *Figure 3f*). Strikingly, the allelic effects of these 9 QTL were undetectable in the presence of the *A. thaliana AP1* genomic locus (*Table 1*). This epistasis was readily observed in the distribution of petal number between plants homozygous for the *A. thaliana AP1* transgene, which had four petals, and plants that lacked the transgene, which had variable petal number (*Figure 3g*). Thus, *A. thaliana AP1* canalized *C. hirsuta* petal number by masking the phenotypic effects of at least 9 QTL.

## Changes in *AP1* expression contribute to species-specific petal number

Our findings suggest that the *AP1* genes from *A. thaliana* and *C. hirsuta* may have a differential ability to confer four petals. To test whether or not *C. hirsuta AP1* could fully complement the function of *A. thaliana AP1*, we introduced a *ChAP1::ChAP1:GFP* transgene into an *ap1* mutant background in *A. thaliana*. Rather than restoring four petals like *ap1-12; AtAP1::AtAP1:GFP* (*Figure 3a*), we found that the distribution of petal number remained variable in *ap1-12; ChAP1::ChAP1:GFP* flowers (homogeneity of variance accepted by Levene's test, p=0.282; *Figure 4a,b*), mimicking the variable petal number found in *C. hirsuta*. Petal number varied between zero and four, and the average petal number was significantly lower in *ap1* plants expressing the *AP1* genomic clone from *C. hirsuta* rather than *A. thaliana* (*Figures 3a* and *4a,b*, Mann-Whitney *U* test p=2.08e-07). In contrast, *ChAP1:: ChAP1:GFP* expression was sufficient to reduce the ectopic flowers in *ap1* mutants (*Figure 4b*), indicating that *AP1* divergence between *C. hirsuta* and *A. thaliana* affected petal number independently of floral meristem identity. Therefore, the results of these gene swaps indicate that *C. hirsuta AP1* has a reduced ability to promote four petals when compared to *A. thaliana AP1*.

Next, we considered the relative contributions of regulatory and coding sequences to this species-specific difference in *AP1* function. To address this question, we expressed endogenous and chimeric versions of *A. thaliana* and *C. hirsuta AP1*, swapping the promoter and coding sequences, in an *ap1* mutant background in *A. thaliana*. We found that petal number was significantly elevated by the *A. thaliana AP1* promoter, compared to the *C. hirsuta AP1* promoter, irrespective of the *AP1* coding sequence driven by these promoters (*Figure 4c*). Whereas all constructs functioned equivalently to rescue the ectopic flowers found in *ap1* mutants (*Figure 4—figure supplement 1*). Therefore, functional differences in *AP1* that are responsible for petal number variation between *A. thaliana* and *C. hirsuta*, are more likely to reside in regulatory regions of the gene rather than coding sequences.

## Species-specific *AP1* expression

Since upstream regulatory regions contributed to *AP1* divergence, we investigated whether expression differed between *C. hirsuta* and *A. thaliana AP1*. We reasoned that the reduced function of *C. hirsuta AP1* to promote four petals may reflect reduced levels of gene expression. To test this prediction, we compared *AP1* expression between *C. hirsuta* lines with matched copy numbers of either *AtAP1::AtAP1:GFP* or *ChAP1::ChAP1:GFP* transgenes, and found that expression levels were significantly lower in floral tissues of *ChAP1::ChAP1:GFP* than *AtAP1::AtAP1:GFP* lines (*Figure 4d*). To visualize the spatiotemporal dynamics of expression, we localized AP1::AP1:GFP fusion proteins from each species in the four-dimensional context of the growing *C. hirsuta* flower (*Figure 4—figure supplement 2*). In stage four floral buds, we observed *A. thaliana* AP1::AP1:GFP nuclear signal in the sepal whorl and on the flanks of the floral meristem, in the small regions where petals initiate in whorl two (*Figure 4e*). In contrast to this, *C. hirsuta* AP1::AP1:GFP was essentially restricted to the sepal whorl throughout stages 4 and 5 (*Figure 4f*, *Figure 4—figure supplement 2*). Using top-view, two-dimensional snapshots of these curved surface images, we measured how far the AP1::AP1:GFP signal extended into whorl two in each transgenic line (*Figure 4—figure supplements 3–5*). We found that the expression of *A. thaliana* AP1::AP1:GFP extended significantly further than *C. hirsuta* AP1::AP1:GFP (approximately 5 µm, *Figure 4g*). Moreover, we found no significant change in size or geometry between flowers expressing either the *A. thaliana* or *C. hirsuta AP1* genomic constructs (*Figure 4g*, *Figure 4—figure supplement 6*). This contrasts with the changes in growth and maturation of floral buds that are associated with the regulation of *C. hirsuta* petal number by seasonal changes in temperature (*McKim et al., 2017*). Therefore, the expression domain of *ChAP1::ChAP1: GFP* is reduced compared to *AtAP1::AtAP1:GFP*, comprising fewer cells in the petal whorl on the flanks of the floral meristem in *C. hirsuta*.

By co-localizing the expression of both *AtAP1::AtAP1:GFP* and *ChAP1::ChAP1:RFP* in stage 4 flowers of *C. hirsuta* and *A. thaliana*, we also found that *C. hirsuta* AP1::AP1:RFP expression is enriched in the regions between sepals (*Figure 4—figure supplement 7*). Since peaks of auxin activity are displaced away from the petal whorl to the region between sepals in *C. hirsuta* flowers (*Figure 2d*), and since distortions of this region have been shown to influence petal initiation in *A. thaliana* (*Baker et al., 2005*; *Lampugnani et al., 2012*; *Lampugnani et al., 2013*; *Laufs et al., 2004*;

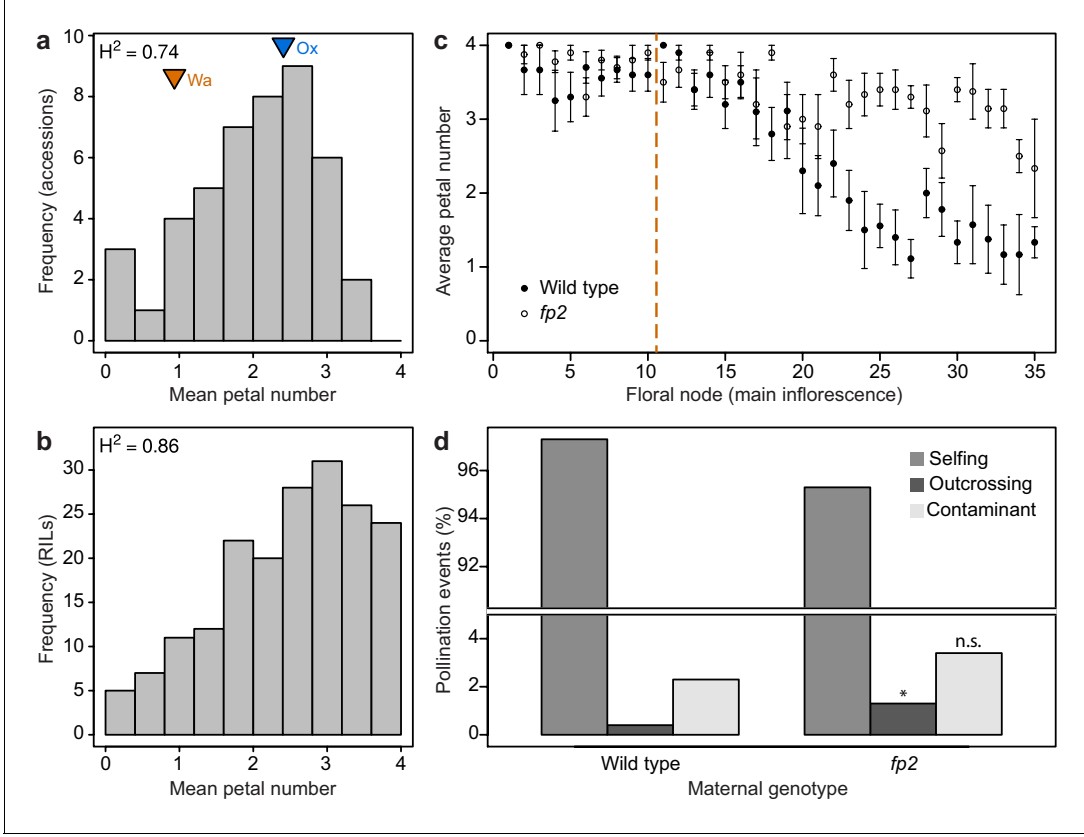

**Figure 5.** Petal number distributions differ between natural and experimental populations of *C. hirsuta* and outcrossing frequency associates with petal number. (**a–b**) Distributions of *C. hirsuta* petal number in 45 natural accessions (**a**) and a population of RILs derived from Ox and Wa accessions (**b**, reproduced from (*Pieper et al., 2016*)). Mean petal number of Ox and Wa are indicated in (**a**). (**c**) Average petal number (±s.e.m.) at every floral node in homozygous wild-type and *fp2* plants (genotyped at SNP:2:2905982) in field conditions (*n* = 10 plants from each genotype). Flowers 1–10 were removed from every plant (indicated by dashed line) since petal number in these flowers did not differ significantly between genotypes (p>0.05, Mann-Whitney *U* test). Seeds produced from remaining flowers on the main inflorescence were harvested. (**d**) Progeny of 10 wild-type and 10 *fp2* mothers were genotyped at SNP:2:2905982 to determine their paternity (*n* = 1703 wild-type and 1610 *fp2* seedlings). Pollination events were considered as selfing when the genotype of the progeny corresponded to the maternal genotype; outcrossing when the genotype of the progeny was heterozygous; and contaminant when the genotype of the progeny corresponded to the other parental genotype. These were likely seed contaminants from the outside of collection bags. Rates of outcrossing were significantly different between genotypes, p<0.05, whereas rates of contaminations were not p>0.05 (Chi-square test with Yates' continuity correction). Moreover, outcrossing and contaminations per parent plant were not correlated (r² = 0.025), suggesting that they are independent events.

DOI: https://doi.org/10.7554/eLife.39399.020

The following figure supplements are available for figure 5:

**Figure supplement 1.** Measurement of fitness traits in wild-type (Ox) and *AtAP1::AtAP1:GFP* plants.

DOI: https://doi.org/10.7554/eLife.39399.021

**Figure supplement 2.** Field experiment to paternity-test progeny of *C. hirsuta* wild-type and *four petals2* (*fp2*) genotypes.

DOI: https://doi.org/10.7554/eLife.39399.022

*Mallory et al., 2004*), the enrichment of *C. hirsuta* AP1 in this domain might be relevant for petal number variation in *C. hirsuta*.

## Maintaining variable petal number in *C. hirsuta*

Our findings suggest that petal number is a robust phenotype in *A. thaliana* that became decanalized in *C. hirsuta*, such that *AP1* divergence allowed previously cryptic loci to quantitatively affect petal number. Allelic variation at these QTL maintains the distribution of petal numbers found among natural accessions of *C. hirsuta* (*Pieper et al., 2016*). A striking feature of this distribution is the absence of natural accessions with four petals (*Figure 5a*). Moreover, few accessions have an

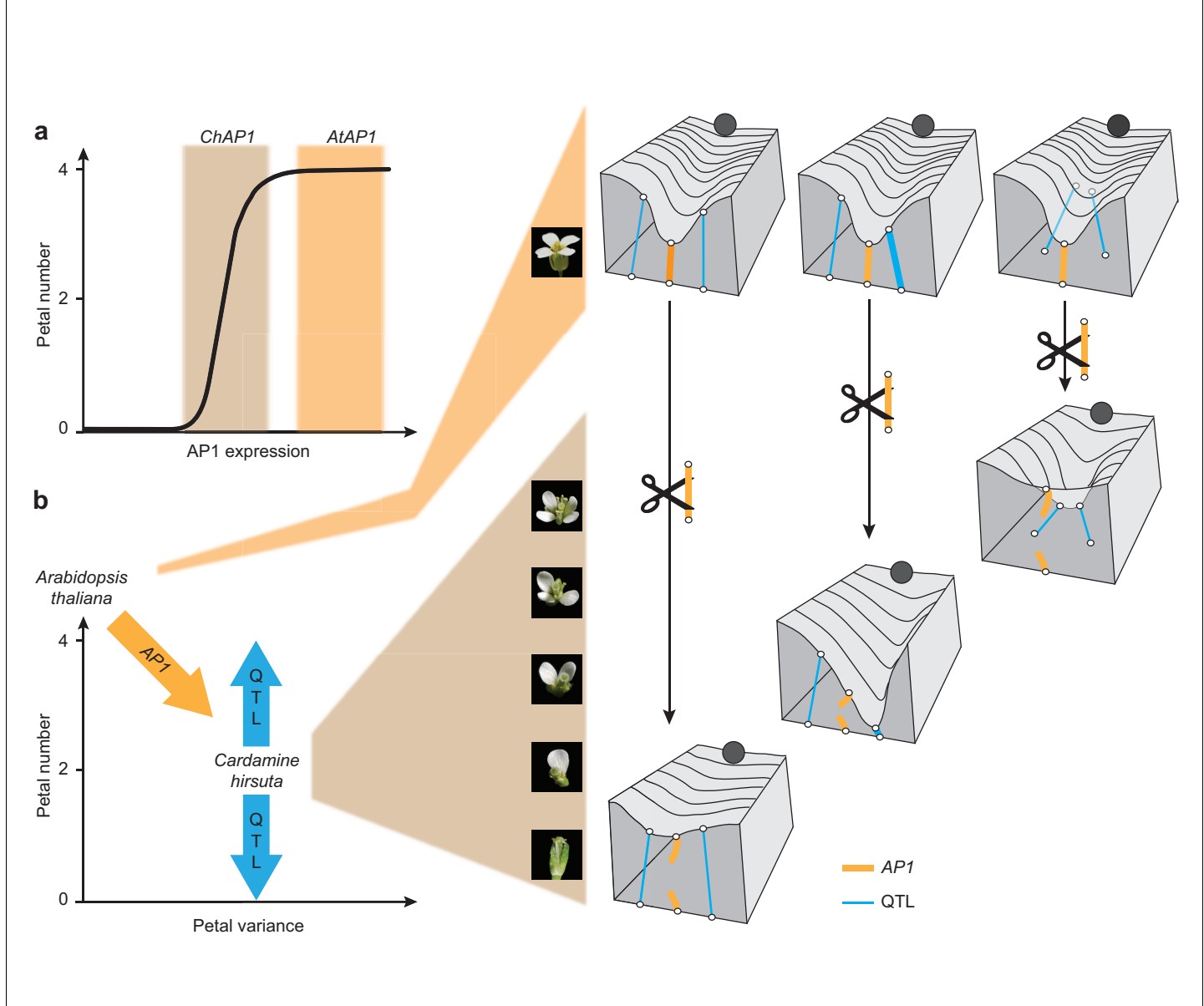

**Figure 6.** Cartoons depicting the proposed role of *AP1* in phenotypic buffering and canalization. (a) Non-linear relationship between *AP1* expression and petal number (black line). The range of *AtAP1* expression (orange) is within a zone of high phenotypic robustness, while the range of *ChAP1* expression (brown) is outside of this robust zone, such that petal number is sensitive to perturbations. (b) Left: decanalization of petal number in *C. hirsuta* from an invariant phenotype of four petals, typified by *A. thaliana*. *AP1* divergence (orange arrow) allowed phenotypic expression of QTL (blue arrows) in the *C. hirsuta* genome to quantitatively affect petal number. Right: cartoons of Waddington's landscape depicting petal number as a canalized phenotype in *A. thaliana* (ball rolls down path of least resistance shaped by canals in the landscape); this landscape is underpinned by a genetic network including *AP1* (orange rope) and QTL (blue ropes). Decanalization of petal number in *C. hirsuta* involved regulatory changes in *AP1* (cut orange rope) that relaxed its epistasis over QTL that cause petal number to vary (deformations in the landscape). Based on (*Waddington, 1957*; *Paaby and Rockman, 2014*).

DOI: https://doi.org/10.7554/eLife.39399.023

average petal number approaching four (*Figure 5a*). To explore the genetic basis of this phenotypic distribution, we constructed experimental populations from founder accessions with phenotypes close to the mean (arrowheads, *Figure 5a*). These recombinant inbred populations contained a high frequency of lines with high average petal numbers, including four petals (*Figure 5b*, and other examples in (*Pieper et al., 2016*)). This shows that there is sufficient standing genetic variation to produce phenotypes through recombination that are not observed in natural accessions.

To explore this finding, we reasoned that producing four-petalled flowers could incur an energetic cost. However, in standard growth conditions, we could not detect any difference in seed output between *C. hirsuta* wild-type and *AtAP1::AtAP1:GFP* plants that differed in petal number (*Figure 5—figure supplement 1*). Alternatively, variable petal number could be the cause or consequence of variation in other trait(s) that contribute to maintaining petal number within its variable range. Given that petals are responsible for the opening of most flowers (*van Doorn and Van Meeteren, 2003*), thus allowing cross-pollination, we tested whether petal number was associated with outcrossing rate in *C. hirsuta*. Using field conditions to grow two genotypes that differed significantly in petal number ($2.65 \pm 0.08$ vs $3.45 \pm 0.05$, Mann-Whitney *U* test $p < 0.001$; *Figure 5c*, *Figure 5—figure supplement 2*), we paternity-tested the progeny of these plants and found a significantly higher outcrossing rate in the genotype with higher petal number (Chi-square test $p < 0.05$, Monte-Carlo permutation test $p < 0.001$, *Figure 5d*, *Figure 5—figure supplement 2*). Therefore, outcrossing frequency is associated with petal number in *C. hirsuta*.

## Discussion

In this study, we identified *AP1* as a gene of major effect in the evolutionary transition from a robust phenotype of four petals, typified by *A. thaliana*, to a variable petal number in *C. hirsuta*. Despite this large effect, *AP1* polymorphisms do not contribute directly to within-species variation in *C. hirsuta* petal number. Instead, the decanalization of petal number in *C. hirsuta* involved regulatory changes in *AP1* that relaxed its epistasis over alleles that cause petal number to vary. Therefore, our results suggest that *AP1* divergence likely exposed cryptic genetic variation in *C. hirsuta* that contributes directly to maintaining variable petal number within natural accessions of *C. hirsuta*.

AP1 is an important regulator of *A. thaliana* flower development, acting early to initiate flowers and later to specify floral organs (*Bowman et al., 1993*; *Irish and Sussex, 1990*; *Mandel et al., 1992*). It functions predominantly as a transcriptional repressor during floral initiation and later as an activator during sepal and petal initiation, and has been described as a 'true hub' in the gene regulatory network that initiates flower development (*Kaufmann et al., 2010*). We found that *AP1* genes from *A. thaliana* and *C. hirsuta* diverged in their ability to specify an invariant number of four petals. This difference mapped broadly to upstream regulatory rather than coding regions of *AP1*, and did not alter the function of *AP1* in flower initiation or sepal specification; suggesting that regulatory sequence divergence can disable specific linkages in gene regulatory networks while maintaining pleiotropic functions in other tissues (*Rebeiz and Tsiantis, 2017*).

We showed that differences in *AP1* expression are associated with the functional divergence in *AP1* between *A. thaliana* and *C. hirsuta*. Specifically, *C. hirsuta AP1* transcript levels are reduced in floral tissues and protein abundance is reduced in the small regions of stage four flowers where petals initiate. Robustness to variation in developmental systems is often a consequence of nonlinear dose-response curves between gene activity and phenotype (*Félix and Barkoulas, 2015*). For example, vulva development in *Caenorhabditis elegans* is robust to genetic variation in *lin-3* expression, such that 15–50 *lin-3* mRNA molecules defines a robust range that allows wild-type cell fate patterning, bounded by two thresholds, beyond which the phenotype varies (*Barkoulas et al., 2013*). We propose that the expression of *A. thaliana AP1* defines a robust range where petal number is buffered against perturbations (*Figure 6a*). *C. hirsuta AP1* is expressed below this threshold, such that petal number varies in response to genetic, environmental, and stochastic perturbations (*Figure 6a*) (*McKim et al., 2017*; *Monniaux et al., 2016*; *Pieper et al., 2016*).

Waddington's model of canalization invokes a similar concept of phenotypic buffering against natural variation. In his classical metaphor of marbles rolling down canals, he depicted the surface of the landscape being pulled down by guy ropes and fastened to anchors that represented genes (*Waddington, 1957*). Like this, the genetic underpinnings of the landscape may vary, but produce a consistent phenotype (e.g. *A. thaliana* petal number, *Figure 6b*) (*Paaby and Rockman, 2014*). Petal number is invariant among natural accessions of *A. thaliana*, reflecting its robustness to genetic variation. However, we could change *A. thaliana* petal number to variable by complementing *ap1* mutants with the *AP1* genomic locus from *C. hirsuta*. These flowers mimicked the variable petal number found in *C. hirsuta*, suggesting that there may be variability in the gene regulatory network controlling petal number in *A. thaliana* that is hidden beneath the uniformity of wild-type development.

In *C. hirsuta*, petal number is decanalized, such that natural genetic variation deforms Waddington's landscape, producing a variable phenotype (*Figure 6b*). We mapped this natural variation to specific QTL in the *C. hirsuta* genome and showed that introducing the *A. thaliana AP1* locus masked the effects of all QTL. Therefore, petal number was effectively canalized in *C. hirsuta* via epistasis of *A. thaliana AP1* over existing QTL. Based on these results, we propose that *AP1* divergence perturbed the genetic equilibrium in *C. hirsuta* that confers petal number robustness. Given the large effect of the *A. thaliana AP1* locus on petal number, it is interesting that endogenous *AP1* polymorphisms do not contribute directly to petal number variation in *C. hirsuta*. Rather, it is the ability of *AP1* to exert epistasis over other loci that diverged between *A. thaliana* and *C. hirsuta* (*Figure 6b*). We propose that this change in genetic interactions was the likely mechanism by which cryptic variation was exposed in *C. hirsuta*, contributing to the evolution of variable petal number.

# Materials and methods

**Key resources table**

| Reagent type (species) or resource | Designation | Source or reference | Identifiers | Additional information |
|---|---|---|---|---|
| Gene (*Cardamine hirsuta*) | AP1 | *Gan et al. (2016)* | CARHR062020 | |
| Gene (*C. hirsuta*) | PTL | *Gan et al. (2016)* | CARHR209620 | |
| Gene (*C. hirsuta*) | AG | *Gan et al. (2016)* | CARHR225900 | |
| Gene (*C. hirsuta*) | Clathrin/AP2M | *Gan et al. (2016)* | CARHR174880 | |
| Biological sample (*C. hirsuta*) | Ox | *Hay and Tsiantis (2006)* | herbarium specimen voucher Hay 1 (OXF) | |
| Biological sample (*Arabidopsis thaliana*) | Col-0 | | CS60000 | |
| Genetic reagent (*A.thaliana*) | pDR5rev::3XVENUS-N7 | *Heisler et al., 2005* | | |
| Genetic reagent (*C. hirsuta*) | pDR5rev::3XVENUS-N7 | *Barkoulas et al., 2008* | | |
| Genetic reagent (*C. hirsuta*) | DR5-v2::3xVenus | *Liao et al., 2015* | | |
| Genetic reagent (*A.thaliana*) | ap1-12 | N6232 | | |
| Genetic reagent (*A.thaliana*) | ptl-1 | N276 | | |
| Genetic reagent (*C. hirsuta*) | ag-1 | this paper | | EMS mutant |
| Genetic reagent (*C. hirsuta*) | fp2 | *Pieper et al. (2016)* | | |
| Genetic reagent (*A.thaliana*) | AtPTL::AtPTL:YFP | this paper | | 2.9 kb genomic sequence of PTL |
| Genetic reagent (*A.thaliana*) | AtPTL::AtPTL:YFP; ptl-1 | this paper | | 2.9 kb genomic sequence of PTL |
| Genetic reagent (*C. hirsuta*) | AtPTL::AtPTL:YFP | this paper | | 2.9 kb genomic sequence of PTL |
| Genetic reagent (*A.thaliana*) | AtAP1::AtAP1:GFP | *Urbanus et al., 2009* | | |
| Genetic reagent (*C. hirsuta*) | AtAP1::AtAP1:GFP | *Urbanus et al., 2009* | | |
| Genetic reagent (*C. hirsuta*) | ChAP1::ChAP1:GFP | *Monniaux et al., 2017* | | |

*Continued on next page*

*Continued*

| Reagent type (species) or resource | Designation | Source or reference | Identifiers | Additional information |
|---|---|---|---|---|
| Genetic reagent (*A. thaliana*) | ChAP1::ChAP1:GFP; ap1-12 | *Monniaux et al., 2017* | | |
| Genetic reagent (*A. thaliana*) | AtAP1::AtAP1:GFP; ap1-12 | *Urbanus et al., 2009* | | |
| Genetic reagent (*A. thaliana*) | ChAP1::ChAP1:RFP; AtAP1::AtAP1:GFP | this paper | | 6.6 kb genomic sequence of AP1 |
| Genetic reagent (*C. hirsuta*) | ChAP1::ChAP1:RFP; AtAP1::AtAP1:GFP | this paper | | 6.6 kb genomic sequence of AP1 |
| Genetic reagent (*A. thaliana*) | pAtAP1::AtAP1; ap1-12 | this paper | | 2.9 kb promoter sequence of AP1 driving AP1 cDNA |
| Genetic reagent (*A. thaliana*) | pAtAP1::ChAP1; ap1-12 | this paper | | 2.9 kb promoter sequence of AP1 driving AP1 cDNA |
| Genetic reagent (*A. thaliana*) | pChAP1::ChAP1; ap1-12 | this paper | | 2.9 kb promoter sequence of AP1 driving AP1 cDNA |
| Genetic reagent (*A. thaliana*) | pChAP1::AtAP1; ap1-12 | this paper | | 2.9 kb promoter sequence of AP1 driving AP1 cDNA |
| Genetic reagent (*C. hirsuta*) | Ox gAtAP1-GFP × Nz F2 | this paper | | 312 individuals used for QTL analysis |
| Software | MorphoGraphX | *Barbier de Reuille et al., 2015* | | |

## Accession numbers and plant material

The wild-type genotype in *C. hirsuta* is the reference Oxford (Ox) accession, herbarium specimen voucher Hay 1 (OXF) (*Hay and Tsiantis, 2006*), and in *A. thaliana*, the reference Col-0 accession. *DR5::VENUS* transgenic lines in *A. thaliana* (*Heisler et al., 2005*) and *C. hirsuta* (*Barkoulas et al., 2008*) have been described previously. NASC accession numbers for *A. thaliana* mutants: *ap1-12* (N6232), *ptl-1* (N276). Additional *A. thaliana* and *C. hirsuta* accessions have been described previously (*1001 Genomes Consortium. Electronic address: magnus.nordborg@gmi.oeaw.ac.at and 1001 Genomes Consortium, 2016*; *Hay et al., 2014*). *C. hirsuta* genome assembly gene identifiers: *ChAP1* (CARHR062020), *ChPTL* (CARHR209620), *ChAG* (CARHR225900), *Clathrin/AP2M* (CARHR174880) (*Gan et al., 2016*).

## Plant growth conditions and petal number scoring

All plants were grown in long day conditions unless otherwise stated. Greenhouse: 16 hr light (22°C), 8 hr dark (20°C); controlled environment room long days: 16 hr light (21°C), 8 hr dark (20°C) and short days: 10 hr light (21°C), 14 hr dark (21°C); growth cabinet short days: 8 hr light (22°C), 16 hr dark (20°C). Petal number was generally scored on 10–15 plants from each genotype, except when scoring the T1 generation of transgenic lines. Flowers were scored every second day by removing them from the plant with tweezers, and observing them with a head band magnifier or stereo microscope. For the *C. hirsuta ag* mutant, flowers from four plants were removed every second day and scored with a binocular loop. Whorl one was removed to allow scoring of whorl two organs, which were then removed to score whorl three organs.

## EMS mutagenesis and *ag* mutant isolation

*C. hirsuta* Ox seeds were mutagenized by agitation with ethyl methanesulfonate (EMS, Sigma), sown on soil, harvested as pools of five M1 plants, and M2 plants were screened for floral phenotypes as described previously (*Pieper et al., 2016*). The *four petals 2* (*fp2*) (*Pieper et al., 2016*) and *agamous* (*ag-1*) mutants were isolated and backcrossed twice to Ox. The *ag-1* sequence bears a C to T single nucleotide change predicted to convert a Gln residue to a stop codon and produce a truncated 33 AA protein. Expressing an *AtAG:GFP* translational fusion (gift from G. Angenent (*Urbanus et al., 2009*)) complemented the *C. hirsuta ag-1* mutant phenotype.

## Quantitative RT-PCR (qPCR) and transgene copy number determination

Five and ten flowers from one plant of *C. hirsuta* wild-type and *ag-1*, respectively, were pooled to generate one biological replicate for RNA extraction. Three biological replicates were generated per genotype. For quantification of *AP1* expression levels in *C. hirsuta AtAP1::AtAP1:GFP* and *ChAP1::ChAP1:GFP* lines, a secondary inflorescence from 29 and 24 plants from five independent lines, respectively, was used for RNA extraction, together with three wild-type biological replicates. For these plants, transgene copy number was determined from genomic DNA by a Taqman qPCR assay using the Hygromycin resistance gene (IDna Genetics, Norwich, UK). RNA was extracted using Spectrum Plant Total RNA kit (Sigma) and DNA was digested by on-column DNase I digestion (Sigma). Reverse Transcription was performed with SuperScript III Reverse Transcriptase (ThermoFisher Scientific) using 1 µg of RNA template. Quantitative PCR was performed with the Power SYBR Green Master Mix (ThermoFisher Scientific) with the following primers: AP1-qPCR-F (5'- CCAGAGGCATTA TCTTGGGGAAGACTTG) and AP1-qPCR-R (5'- GCTCATTGATGGACTCGTACATAAGTTGGT) to amplify either *ChAP1* or *AtAP1*, and Clathrin-qPCR-F (5'- TCGATTGCTTGGTTTGGAAGATAAGA) and Clathrin-qPCR-R (5'- TTCTCTCCCATTGTTGAGATCAACTC) to amplify the reference gene *Clathrin/AP2M*. Expression was calculated with the ΔΔCt method (**Pfaffl, 2001**), normalized against the reference gene, and expressed relative to wild-type levels.

## Transgenic plant construction

For the *AtPTL::AtPTL:YFP* construct, a 2.8 kb *PTL* promoter up to the second exon, driving functional *PTL* expression (**Lampugnani et al., 2012**), was amplified with primers pPTL-F (5'-ATATA TTGAGAAGAGATTAAAAACTTAG) and pPTL-R (5'-GTATCCATGTTCCTCGGACA) from Col-0 genomic DNA and cloned into the multiSite Gateway donor vector pDONR-P4-P1R. The full 2.9 kb genomic sequence of *PTL* was amplified with primers gPTL-F (5'-ATGGATCAAGATCAGCATC) and gPTL-R (5'-CTGATTCTCTTCTTTACTGAGC) from Col-0 genomic DNA and cloned into the multiSite Gateway donor vector pDONR-221. The *YFP* coding sequence was cloned into the multiSite Gateway donor vector pDONR-P2R-P3. The *AtPTL::AtPTL:YFP* construct was created by recombining together the three previous vectors into the pGII-0229 destination vector, as described in the MultiSite Gateway manual (Thermo Fisher Scientific). Eight and nineteen independent lines of *AtPTL:: AtPTL:YFP* were generated in *C. hirsuta* wild type and *A. thaliana ptl-1* respectively. Petal number was scored on all independent lines in the T1 generation, together with *C. hirsuta* wild type and *A. thaliana ptl-1*.

Twenty independent lines of *AtAP1::AtAP1:GFP* (gift from G. Angenent (**Urbanus et al., 2009**)) were generated in *C. hirsuta* wild-type and *A. thaliana ap1-12* backgrounds. This translational fusion contains a 6.6 kb genomic fragment of *A. thaliana AP1* including 3 kb of regulatory sequence upstream of the translational start. *ChAP1::ChAP1:GFP* was previously described (**Monniaux et al., 2017**) and contains a comparable 6.6 kb genomic fragment of *C. hirsuta AP1*. Ten independent lines of *ChAP1::ChAP1:GFP* were generated in *C. hirsuta* wild type and *A. thaliana ap1-12*. For all the *AP1*-related lines, petal number was scored on 2 to 5 single-insertion homozygous T3 lines, together with wild-type *C. hirsuta* and *A. thaliana ap1-12*.

*ChAP1::ChAP1:RFP* was constructed in the modified destination vector pB7RWG2 (gift from M. Kater (**Gregis et al., 2009**)) by recombining the same *C. hirsuta AP1* genomic fragment used above. Six independent lines were generated in *A. thaliana AtAP1::AtAP1:GFP* for co-localisation studies. Four independent lines were generated in *C. hirsuta* and selected for strong expression in the third generation. Homozygous plants were crossed to an *AtAP1::AtAP1:GFP* strong expressing line for co-localisation studies.

The *pAtAP1::AtAP1*, *pAtAP1::ChAP1*, *pChAP1::AtAP1* and *pChAP1::ChAP1* constructs were generated by three-fragment multi-site Gateway in the pGII-0227 destination vector. The *AtAP1* and *ChAP1* promoters contain 2.9 kb upstream of the start codon and were amplified with primers pAtAP1-F (5'- CGAACGTGGTGGTTAGAAGA) and pAtAP1-R (5'-TTTTGATCCTTTTTTAAGAAAC TT), and primers pChAP1-F (5'-CATATAGCTTGGATCATGCTC) and pChAP1-R (5'-TTTGATCCTA TTTTGAGAAACTTCTT) respectively. Ten independent lines, with five plants per line, were scored for petal and ectopic flower number together with *A. thaliana* wild type and *ap1-12*. Lines with a clear *ap1-12* phenotype were considered not to be complemented by the transgene and were removed from the analysis.

The *DR5-v2::3xVENUS* plasmid was a gift from Dolf Weijers (*Liao et al., 2015*). Eleven independent insertion lines were generated in *C. hirsuta* wild type. All lines were checked for expression in the T1 generation and two representative lines were selected to image by time-lapse CLSM in the T2 generation.

All binary vectors were transformed into *C. hirsuta* or *A. thaliana* by *Agrobacterium tumefaciens* (strains GV3101 or C58)-mediated floral dip.

## Scanning electron microscopy (SEM)

Shoot apices were induced to flower by a shift from short to long day conditions and fixed in FAA, post-fixed in osmium tetroxide, dehydrated, critical point dried and dissected before coating with gold/palladium for viewing in a JSM-5510 microscope (JEOL). Floral primordia were staged according to (*Smyth et al., 1990*).

## Confocal laser scanning microscopy (CLSM) and quantitative image analysis

Time-lapse imaging was performed using 4–5 week old plants grown on soil in long day conditions. The inflorescence was cut and flowers were dissected off to uncover young floral primordia at the shoot apex. Dissected shoots (around 0.5 cm long sections) were transferred to ½ MS medium supplemented with 1.5% plant agar, 1% sucrose and 0.1% Plant Preservative Mixture (Plant Cell Technology). To outline cells, samples were stained with 0.1% propidium iodide (PI, Sigma) for 2 – 5 min before each observation. Floral primordia were immersed in water and imaged from the top at 24 hr intervals. Confocal imaging was performed using a Leica SP8 up-right confocal microscope equipped with a long working-distance water immersion objective (L 40x/0.8 W) (Leica) and HyD hybrid detectors (Leica). Excitation was achieved using an argon laser with 514 nm for VENUS and PI. Images were collected at 526 – 545 nm for VENUS, and 600 – 660 nm for PI. Between imaging, samples were transferred to a growth cabinet and cultured in vitro in standard long day conditions at 20°C. Confocal image stacks of time-lapse series were analyzed using MorphoGraphX software (*Barbier de Reuille et al., 2015*; *Kierzkowski et al., 2012*). The outer 10 Full datasets of *A. thaliana* and *C. hirsuta* time-lapse series used to track growth and *DR5::VENUS* and *DR5v2:: VENUS* expression shown in *Figure 2—figure supplement 1*. To measure physical boundary size and size of AP1-GFP expression, the epidermal (2 to 5 μm) GFP signal was projected on the surface of the sample with MorphoGraphX. Top-view snapshots of the flower meristem with GFP-projected signal were acquired and subsequently analyzed with FiJi (*Schindelin et al., 2012*) as described in *Figure 4—figure supplement 3*.

## Histology and in situ hybridization

Shoot apices were induced to flower by a shift from short to long day conditions. Digoxigenin-labeled antisense RNA probes to *C. hirsuta ChPTL* were generated by mixing together three synthetic probes covering the whole *ChPTL* cDNA (GenScript HK Limited, USA). 8 – 10 μm inflorescence cross-sections were fixed, embedded in paraffin and hybridized with the *ChPTL* probe as previously described (*Vlad et al., 2014*). The signal was observed and images were acquired with a Zeiss AxioImager.M2 light microscope equipped with an Axiocam HR color camera. To cover the entire hybridization pattern in depth, images of two consecutive sections were registered and minimal projections generated using the image processing package Fiji (*Schindelin et al., 2012*). Cropping, gamma and colour correction were done using Photoshop CS5 and performed on entire images only. For semi-thin sections, apices were fixed in 2.5% glutaraldehyde in phosphate buffer, dehydrated, step-wise infiltrated with and embedded in TAAB Low Viscosity resin (TAAB) and 1.5 μm sections were stained with 0.05% toluidine blue.

## Quantitative trait locus (QTL) analysis

QTL analysis of petal number was performed on a *C. hirsuta* F2 population derived from a cross between an *AtAP1::AtAP1:GFP* transgenic line in the Ox accession and the Nz accession. Petal number was quantified in 312 individuals that were pre-screened by PCR amplification of the *GFP* sequence such that approximately 1/4 of the plants were wild-type. The first 25 flowers on each plant were removed on the day they opened and petal number was counted using a dissecting

microscope. The 312 F2 plants were genotyped with 155 Sequenom markers (Welcome Trust Center for Human Genetics, High Throughput Genomics, Oxford, UK) designed to cover the whole genome according to an early version of the *C. hirsuta* genome assembly (*Gan et al., 2016*), and a dCAPS marker was generated for the *C. hirsuta AP1* locus using primers AP1cisF1 (5'-TCCCTAAAACCGC TCTTAGC) and AP1cisR1 (5'-AGAGAGATAAAGAAGAGTTCAGGC) and the restriction enzyme *Alu*I. The genetic map was made using JoinMap 4 (*Van Ooijen, 2006*), including the genotype for *AtAP1::AtAP1:GFP* as a dominant marker to determine the location of the transgene, and had a total length of 910 centiMorgans in eight linkage groups. QTL analyses were performed with Genstat 13th Edition (VSN International, Hemel Hempstead, UK) using all 312 F2 plants. Genetic predictors were calculated with a maximum distance of 2 cM between them from the molecular marker data and the genetic map. Average petal number per F2 plant was used as a phenotype for QTL analysis. Simple interval mapping and composite interval mapping were performed. The latter procedure was repeated several times while adding and/or removing cofactors until no further improvement could be made. The resulting set of 10 cofactors was used in a final QTL model to estimate QTL effects. A model with nine cofactors, when excluding the *AtAP1::AtAP1:*GFP locus, was fitted to data from F2 plants that were either homozygous for the *AtAP1::AtAP1:*GFP transgene or wild-type.

### Field experiment and paternity testing of *C. hirsuta* wild type and *four petals 2* (*fp2*)

Seeds from Ox and *fp2* (*Pieper et al., 2016*) were stratified for 1 week at 4°C and sown on 15.03.2016 on hydrated Jiffy plugs. Seedlings were first grown in a greenhouse without temperature or light control, and later transferred to the field on 13.04.2016. More details on experimental design can be found in *Figure 5* and *Figure 5—figure supplement 2*. Genomic DNA was extracted from parent and progeny plants with Edwards Buffer and isopropanol precipitation and amplified with primers m458 (5'-GCCTAATCTTGCACAACACGAAATCT) and m459 (5'-GATTCTAAAGTTCTG TCAAAAGGAGAAACC**T**GA), designed with dCAPS Finder (http://helix.wustl.edu/dcaps/dcaps. html), to genotype SNP:2:2905982 by dCAPS. PCR was performed with Mango *Taq* polymerase (Bioline) under the following conditions: initial denaturation of 5 min at 95°C, 40 cycles of 30 s at 95°C, 30 s at 56°C and 30 s at 72°C, final extension of 10 min at 72°C. 1/5$^{th}$ volume of the reaction was digested with 2.5 units of *Dde*I (New England Biolabs) for 2 hr at 37°C and migrated on a 3% agarose gel to resolve the uncut 141 bp amplicon for *fp2*, and the two cut fragments of 116 bp and 25 bp for Ox.

### Fitness measurements

40 plants of *C. hirsuta* wild type and a homozygous T4 line of *AtAP1::AtAP1:GFP* were grown in a greenhouse with standard conditions (20°C, long days). 20 of these plants were scored for petal number, and the other 20 were bagged carefully to recover all seeds. A fraction of the seeds was counted and weighed with the seed analyser MARVIN (GTA Sensorik GmbH) and total seed number was estimated by proportionality with the total seed weight.

## Acknowledgements

We thank M Tsiantis for sharing unpublished *C. hirsuta* accessions and for helpful discussions and continuous support throughout this work. We thank P Huijser and R Berndtgen for assistance with *in situ* hybridisation, C Rojas for assistance with field experiments, W Faigl, G Angenent, M Kater, D Weijers and the Arabidopsis Biological Resource Center for materials, J Baker and M Kalda for photography, and S Laurent, M Abraham, and D Bailey for advice. This work was supported by Biotechnology and Biological Sciences Research Council grant BB/H01313X/1 to AH and Human Frontiers Science Program grant RGP0008/2013 to RS. AH was supported by a Royal Society University Research Fellowship and the Max Planck Society W2 Minerva programme, MM and SM by European Molecular Biology Organization Long Term Fellowships, SM by a National Science and Engineering Research Council of Canada Post-Doctoral Fellowship, and DK by Deutsche Forschungsgemeinschaft SFB 680 grant to M Tsiantis.

## Additional information

### Funding

| Funder | Grant reference number | Author |
|---|---|---|
| Biotechnology and Biological Sciences Research Council | BB/H01313X/1 | Angela Hay |
| Human Frontier Science Program | RGP0008/2013 | Richard S Smith |
| Royal Society | University Research Fellowship | Angela Hay |
| Max-Planck-Gesellschaft | W2 Minerva Fellowship | Angela Hay |
| European Molecular Biology Organization | Long Term Fellowship | Marie Monniaux Sarah M McKim |
| Natural Sciences and Engineering Research Council of Canada | Post-Doctoral Fellowship | Sarah M McKim |

The funders had no role in study design, data collection and interpretation, or the decision to submit the work for publication.

### Author contributions

Marie Monniaux, Investigation, Visualization, Writing—original draft, Writing—review and editing; Bjorn Pieper, Investigation, Visualization, Writing—review and editing; Sarah M McKim, Daniel Kierzkowski, Investigation; Anne-Lise Routier-Kierzkowska, Investigation, Visualization; Richard S Smith, Supervision, Funding acquisition; Angela Hay, Conceptualization, Supervision, Funding acquisition, Investigation, Writing—original draft, Project administration, Writing—review and editing

### Author ORCIDs

Marie Monniaux (iD) http://orcid.org/0000-0001-6847-3902
Bjorn Pieper (iD) http://orcid.org/0000-0002-0357-6254
Sarah M McKim (iD) http://orcid.org/0000-0002-8893-9498
Richard S Smith (iD) http://orcid.org/0000-0001-9220-0787
Angela Hay (iD) http://orcid.org/0000-0003-4609-5490

### Decision letter and Author response

Decision letter https://doi.org/10.7554/eLife.39399.028
Author response https://doi.org/10.7554/eLife.39399.029

## Additional files

### Supplementary files
• Transparent reporting form
DOI: https://doi.org/10.7554/eLife.39399.024

### Data availability
All data generated or analysed during this study are included in the manuscript and supporting files.

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
