## [Decision Letter]

Thank you for submitting your article "The role of APETALA1 in petal number robustness" for consideration by *eLife*. Your article has been reviewed by four peer reviewers, and the evaluation has been overseen by a Reviewing Editor and Christian Hardtke as the Senior Editor. The following individuals involved in review of your submission have agreed to reveal their identity: Andrew Hudson (Reviewer #2); Takashi Tsuchimatsu (Reviewer #3).

The reviewers have discussed the reviews with one another and the Reviewing Editor has drafted this decision to help you prepare a revised submission.

Summary:

The genetic control and evolution of phenotypic robustness is an important topic in animal and plant developmental biology. Petal number of *Cardamine hirsuta* varies from 0 to 4 within and between individuals, unlike most of Brassicaceae species, including *Arabidopsis thaliana*, which have stable four-petal flowers. The authors have provided novel insight into the question of phenotypic robustness by identifying regulatory variation in AP1 as the major cause for the loss of petal-number robustness in *Cardamine hirsuta* (compared to *Arabidopsis*), and by demonstrating that ChAP1 appears to have lost the ability to suppress the phenotypic expression of variation at petal-number loci. Interestingly, AP1 itself does not change petal number directly; rather AtAP1 buffers the cryptic genetic variation for petal number, while ChAP1 has lost the ability, decanalizing the polygenic effects of petal number QTLs. The identification of AP1 as a 'canalization factor' is novel and interesting and on this basis the decision was "revise full submission".

However, the consensus among all four reviewers was that there problems with the second part of the paper, where the authors present results of a simulation study and use it as an argument for natural selection maintaining petal-number variability; there are not sufficiently convincing arguments against the default interpretation that petal numbers in *Cardamine* are reduced due to the accumulation of slightly deleterious mutations (in terms of petal-number robustness) that are not or only weakly counter-selected due to their selfing reproductive system. For simulating the populations, the authors assumed that each of the 15 QTL they consider has an equal probability of mutating from an Ox reference to an alternative allele. In their Pieper et al. study they proposed that the Ox allele decreases petal number at 8 of the 15 loci compared to the alternative allele. Thus, at each step in the simulation there would be a roughly equal chance of a mutation increasing or decreasing petal number, depending on which of the loci mutates to an alternative allele. The authors justify this based on their experimental QTL mapping studies, yet it implicitly assumes that all of the detected Ox QTL alleles represent the ancestral states. This (and the corollary of an equal chance of a novel mutation increasing or decreasing petal number) seems rather implausible. An at least equally if not more plausible scenario would be that the different, very distantly related *Cardamine* accessions used for their QTL studies each carry different sets of 'degenerative' mutations reducing petal number, such that the detected petal-number increasing QTL alleles mainly represent the ancestral alleles at these loci still present in the other accessions. This would imply that novel mutations increasing petal number are much rarer than ones decreasing it (even though the former can clearly be found as shown by their *fp2*. Under such a scenario with unequal probabilities for the two types of mutations, many fewer lineages with four petals should accumulate in their simulations, likely obviating the need to invoke natural selection. That mutations reducing petal number can accumulate neutrally as cryptic variation is demonstrated by their ChAP1 transformation into the *Arabidopsis ap1* mutant, resulting in similar levels of petal-number variability as seen in *Cardamine*. Directly testing the plausibility of the simulation would require determining the relative frequencies of petal-number increasing and decreasing novel mutations, which seems unfeasible. Therefore, this simulation and the conclusion of selection maintaining petal-number variability appears highly speculative. In addition, there is ambiguity about what the authors mean by 'selection maintaining variability'. Do they envision that less than four petals are actively selected for? If so, what could be a possible advantage (increased seed production seems unlikely based on their fitness experiment) and why is this seen so very rarely in the many selfing Brassicaceae? Or do they envision selection favoring four petals (e.g. due to the higher outcrossing rates they demonstrate), but this being counteracted by a steady trickle of mutations reducing petal number? Lastly, the statement “Variable petal number is maintained in natural populations of *C. hirsuta* despite

the fact that they harbor sufficient genetic variation to produce flowers with an invariant phenotype of four petals” in the Discussion seems only poorly supported by the authors' data. Their previous QTL mapping used six very distantly related accessions collected from all over the globe, so it is not at all clear that there is sufficient genetic variation within local populations to recreate four-petal genotypes at an appreciable frequency. Further related to this point, the selection regime assumed in this model (selection against high petal number) is somewhat arbitrary, and would not be supported by empirical data, particularly because the cost of high petal number was not detected. To provide context for the model, the authors performed a field experiment, but this experiment detects higher outcrossing rate in four-petal individuals, which would not be directly related the disadvantage of high petal number assumed in the model. It is interesting that the outcrossing rate appeared to be correlated with petal number, but the selective pressure acting on outcrossing rate is quite complex: in addition to what authors discussed, inbreeding depression, transmission advantage, and pollen discounting would be major forces for it (reviewed in detail in Shimizu and Tsuchimatsu, 2015 Ann Rev Eco Evo Syst, for example).

Therefore, this section of the manuscript should be removed.

Essential revisions:

1) In Figure 2, the authors show reconstructed pictures of developing *Arabidopsis* and *Cardamine* flowers expressing DR5-VENUS to claim that there are some defects in DR5-VENUS expression in *Cardamine*. It is difficult to observe these defects in Figure 2B and D. Given that DR5-VENUS expression can be noisy, even in *Arabidopsis*, more pictures of different flower buds would be helpful (including the original stacks or projections and not only the reconstructions). Also, is it possible to quantify the differences in mean number of DR5-VENUS peaks in the 2nd whorl in both species to fully support the claim that C. hirsuta has defects in petal initiation and not emergence?

2) Epistasis of the *Arabidopsis* AP1 transgene over some minor *Cardamine* QTLs (Figure 3G). Is there a way of showing this that expresses the relative probabilities of there being a QTL in the two AP1 genotypes, or at least shows that the power to detect a QTL was the same for both? Table 1 essentially says only that no QTL was detected in the AP1 homozygotes.

3) The authors propose that differences in AP1 expression between species could cause the phenotype. Using qPCR data, they show that the ChAP1 expression is lower than AtAP1 expression in plants carrying 2 or 4 copies, but does that affect the phenotype? The results in Figure 4C are difficult to interpret, due to the high level of variability in petal number for all constructs (even the *ap1* mutant complemented with pAtAP1::AP1, contrary to the equivalent construct in Figure 3A where the distribution appears tightly around 4). But when looking at a double AtAP1 and ChAP1 marker in *Arabidopsis*, the authors also observe changes in expression in the area between sepals (which is not seen in *Cardamine*, based on the picture in Figure 4E and F and in Figure S5). Please discuss these differences. Is more AP1 needed in the second whorl or less AP1 between sepals in the first whorl to promote petal formation? An analysis of the pattern of expression of AtAP1 and ChAP1 in *Arabidopsis* similar to the one performed in *Cardamine* using the double reporter might help. Please also compare these results with the ones in their previous article (Mc Kim et al., 2017), where they showed that changes in growth of the floral buds correlated with the variations in petal number observed in response to seasonal changes in temperature. The analysis in *Cardamine* (Figure 4E to G) seems to suggest that ChAP1 is less expressed in the second whorl than AtAP1, which could cause the phenotype. Do *Cardamine* plants carrying more copies of CdAP1 also produce more petals and is the variability in petal number reduced? This would support the point raised by the authors.

---

## [Author Response]

Summary:The genetic control and evolution of phenotypic robustness is an important topic in animal and plant developmental biology. Petal number of Cardamine hirsuta varies from 0 to 4 within and between individuals, unlike most of Brassicaceae species, including Arabidopsis thaliana, which have stable four-petal flowers. The authors have provided novel insight into the question of phenotypic robustness by identifying regulatory variation in AP1 as the major cause for the loss of petal-number robustness in Cardamine hirsuta (compared to Arabidopsis), and by demonstrating that ChAP1 appears to have lost the ability to suppress the phenotypic expression of variation at petal-number loci. Interestingly, AP1 itself does not change petal number directly; rather AtAP1 buffers the cryptic genetic variation for petal number, while ChAP1 has lost the ability, decanalizing the polygenic effects of petal number QTLs. The identification of AP1 as a 'canalization factor' is novel and interesting and on this basis the decision was "revise full submission".However, the consensus among all 4 reviewers was that there problems with the second part of the paper, where the authors present results of a simulation study and use it as an argument for natural selection maintaining petal-number variability […] Therefore, this section of the manuscript should be removed.

Given this decision, we have removed the simulation experiments from the manuscript and are happy to keep a strong developmental genetics focus to the manuscript. In addition, we have removed the comment about the contribution of natural selection from the Introduction. However, given the public nature of manuscript evaluation correspondence in *eLife*, we would like to respond to the reviewers’ comments about our analysis of standing genetic variation:

Lastly, the statement “Variable petal number is maintained in natural populations of C. hirsuta despite the fact that they harbor sufficient genetic variation to produce flowers with an invariant phenotype of four petals.” seems only poorly supported by the authors' data. Their previous QTL mapping used six very distantly related accessions collected from all over the globe, so it is not at all clear that there is sufficient genetic variation within local populations to recreate four-petal genotypes at an appreciable frequency.

Standing genetic variation is the allelic variation that is currently segregating, as opposed to alleles that appear by new mutation events. Our question is whether new mutations, such as *fp2*, are required for *C. hirsuta* flowers to produce high average petal number, or whether natural allelic variation in *C. hirsuta* is sufficient to produce this phenotype. Crossing accessions and analysing the distribution of phenotypes produced by segregating alleles is a standard method to assess the extent of phenotypic variation produced by standing genetic variation. Moreover, we crossed accessions close to the mean of the phenotypic distribution rather than those at opposite extremes, to avoid bias.

The reviewers make assumptions about the scale at which genetic variation is relevant for the phenotype – “*it is not at all clear that there is sufficient genetic variation within local populations to recreate four-petal genotypes*”. Recent work in *Arabidopsis* has shown large genetic diversity among strains over very short geographic distances (Figure 3D,E in The 1001 Genomes Consortium, 2016. 1,135 Genomes Reveal the Global Pattern of Polymorphism in *Arabidopsis thaliana.* Cell2: 481-91). We have no reason to believe *C. hirsuta* is different in this respect given its similarity to *A. thaliana* in global, and local distribution (Hay et al., 2014). However, investigating these interesting questions requires future, dedicated studies that are well beyond the scope of this current paper.

We have now modified Figure 5 to include the petal distributions of natural and experimental populations and the association between petal number and outcrossing rate. Given that this data addresses the role of genetic variation and trait pleiotropy in maintaining variable petal number, and not the role of natural selection in this process, we hope that the reviewers can now view the data more objectively.

[…] there are not sufficiently convincing arguments against the default interpretation that petal numbers in Cardamine are reduced due to the accumulation of slightly deleterious mutations (in terms of petal-number robustness) that are not or only weakly counter-selected due to their selfing reproductive system. For simulating the populations, the authors assumed that each of the 15 QTL they consider has an equal probability of mutating from an Ox reference to an alternative allele. In their Pieper et al. study they proposed that the Ox allele decreases petal number at 8 of the 15 loci compared to the alternative allele. Thus, at each step in the simulation there would be a roughly equal chance of a mutation increasing or decreasing petal number, depending on which of the loci mutates to an alternative allele. The authors justify this based on their experimental QTL mapping studies, yet it implicitly assumes that all of the detected Ox QTL alleles represent the ancestral states.

The aim of our simulations was to test whether the truncated distribution of average petal numbers observed in natural accessions of *C. hirsuta* could occur when 15 known petal number QTL evolved independently. The lineage with Ox alleles used at the start of the simulation was entirely arbitrary. Our choice merely reflected the fact that Ox had a phenotype closer to the mean petal number of sampled natural accessions than any other founder accession used for QTL mapping.

This (and the corollary of an equal chance of a novel mutation increasing or decreasing petal number) seems rather implausible. An at least equally if not more plausible scenario would be that the different, very distantly related Cardamine accessions used for their QTL studies each carry different sets of 'degenerative' mutations reducing petal number, such that the detected petal-number increasing QTL alleles mainly represent the ancestral alleles at these loci still present in the other accessions.

Here, the reviewers make many assumptions; for example, about geographic distance, genetic diversity, distribution of alleles, ancestry of alleles, and allelic effects being degenerative or beneficial. This is problematic. Many of their assumptions can only be supported or refuted by knowing the molecular basis of these QTL, which is not known. Other assumptions are not borne out by data. For example, the genetic distance between Ox and the other accessions used in previous QTL studies (Pieper et al., 2016) varied greatly, and the number of detected QTL did not depend on this. To give a specific example, the Nz accession, used in the current manuscript, is more closely related to Ox than are the other accessions used in our previous QTL studies, which is why the Ox × Nzgenetic map had only poor coverage of the entire genome (Pieper et al., 2016). Therefore, the assumption that all accessions previously used for QTL analysis are“very distantly related” is not borne out by data.

This would imply that novel mutations increasing petal number are much rarer than ones decreasing it (even though the former can clearly be found as shown by their fp2. Under such a scenario with unequal probabilities for the two types of mutations, many fewer lineages with four petals should accumulate in their simulations, likely obviating the need to invoke natural selection.

The reviewers argue that novel mutations will preferentially decrease petal number. However, this would skew the distribution of petal numbers towards zero when neutral, which does not agree with the truncated normal distribution that we observe in natural accessions. Additionally, a hypothetical optimum petal number close to the observed mean of natural accessions could not account for the truncated tail towards high petal numbers. These inconsistencies between ‘thought experiments’ and observed data are precisely what prompted us to investigate the distribution of petal numbers in an evolving system under different scenarios.

Essential revisions:1) In Figure 2, the authors show reconstructed pictures of developing Arabidopsis and Cardamine flowers expressing DR5-VENUS to claim that there are some defects in DR5-VENUS expression in Cardamine. It is difficult to observe these defects in Figure 2B and D. Given that DR5-VENUS expression can be noisy, even in Arabidopsis, more pictures of different flower buds would be helpful (including the original stacks or projections and not only the reconstructions). Also, is it possible to quantify the differences in mean number of DR5-VENUS peaks in the 2nd whorl in both species to fully support the claim that C. hirsuta has defects in petal initiation and not emergence?

We now present a panel of CLSM stacks for an additional 7 flowers from each species in Figure 2—figure supplement 1 to support the data presented in Figure 2A-D. For this, we decided to repeat the time-lapse analysis for *C. hirsuta* using the *DR5v2::VENUS* construct, as this is reportedly more sensitive and enabled the visualization of previously unobserved weak auxin responses in embryos, seedlings and roots (Liao et al., 2015). We show three consecutive 24-h time points for a representative flower from each species and top and side views of all other flowers at the informative time point for petal initiation. We used this dataset to quantify the percentage of petal initiation sites on the floral meristem where a peak of *DR5-VENUS* or *DR5v2::VENUS* expression is observed: in *A. thaliana*, we observed 83% whereas in *C. hirsuta*, we observed only 36%. We have included these quantifications in the relevant section of the main text and in the legend of Figure 2—figure supplement 1. As the reviewer points out, *DR5-VENUS* peaks are transitory and mark only a few cells at sites of petal initiation amongst the ‘noise’ of expression in other floral organs throughout the flower. For these reasons, it was important that we used live imaging to follow the dynamics of *DR5-VENUS* expression and that we quantified the fluorescence signal on curved surface images extracted from these 3D CLSM stacks. This level of analysis is currently at the leading edge of plant morphodynamics – using quantitative imaging to study the dynamics of gene expression during morphogenesis. By quantifying the difference in second whorl peaks of *DR5v2::VENUS* in *C. hirsuta* (36%) and *DR5-VENUS* in *A. thaliana* (83%), we now present a dataset that more fully supports the claim that *C. hirsuta* has defects in petal initiation.

2) Epistasis of the Arabidopsis AP1 transgene over some minor Cardamine QTLs (Figure 3G). Is there a way of showing this that expresses the relative probabilities of there being a QTL in the two AP1 genotypes, or at least shows that the power to detect a QTL was the same for both? Table 1 essentially says only that no QTL was detected in the AP1 homozygotes.

Our QTL approach made no assumptions about genetic interactions between the *A. thaliana AP1* genomic locus and endogenous *C. hirsuta* loci. We performed QTL analysis on 312 F2 plants from an Ox *gAtAP1-GFP* × Nz F2 mapping population and detected 10 petal number QTL. During this analysis the power to detect QTL played a role and it was high enough to detect even minor effect QTL. The estimated effects of this analysis are shown in Table 1 below for single allele changes from Ox to NZ (for transitions from homozygous Ox to homozygous NZ the additive effect is twice as large, as reported in Table 1 of the main text).

The petal number distributions in Figure 3G show that, owing to its large additive effect, plants homozygous for the *gAtAP1-GFP* transgene (QTL2) have a very narrow distribution of petal number close to the maximum of 4 petals. In fact, the range of average petal numbers in this class is 3.36 – 4 (0.64 petals) with a mean of 3.76 petals. From this, it follows that the sum of the absolute additive effects for homozygous transitions of any QTL controlling this variation cannot be larger than 0.64, which is smaller than the estimated effect of QTL8 alone (1.24 petals for homozygous substitutions).

The sum of absolute additive effects in Table 1 (above) is 2.9 petals, which means that F2 plants homozygous for alleles with positive effect at all QTL would differ by > 4 petals from plants homozygous for alleles with negative effect (2 × 2.9 > 4). Thus, under the naïve assumption that heritability is 1, the detected QTL can largely explain the phenotypic variation in petal number observed in wild-type plants, which ranges from 0 to 4 petals.

However, Ox and NZ alleles are segregating at all of the detected QTL (excluding the *gAtAP1-GFP* locus) in plants homozygous for the *gAtAP1-GFP* transgene, but in these plants the observed range of phenotypes is much smaller than 0 to 4 petals. Therefore, *gAtAP1-GFP* must have an epistatic effect over the other QTL by restricting the effects of these loci. This is exactly what we found when we tested for significance of the allelic effects of the detected QTL in wild-type, and transgenic plants separately (Table 1). The power to detect QTL effects in subsets of the F2 population is likely to be reduced, yet this applies both to the subset of wild-type plants and the subset of transgenic plants.

We illustrate these findings by fitting linear models to petal numbers and the genotype predictors at the QTL in plants with and without the *gAtAP1-GFP* transgene separately (Author response image 1). Note that the regression coefficients (slopes) are not the same in both groups and that these are larger in the absence of the transgene (blue line), typical for an epistatic interaction.

To clarify more precisely how we used a simple QTL analysis to detect epistatic interactions between the *A. thaliana AP1* genomic locus and endogenous *C. hirsuta* loci, we have now written the Materials and methods section for this experiment in more detail.

**Author response image 1. respfig1:** Single marker analysis of petal number QTL in the Ox *gAtAP1-GFP* × Nz F2 mapping population conditional on the presence of *gAtAP1-GFP*. Petal number was regressed on the genotype predictors from the detected QTL in plants with (orange) and without (blue) the *gAtAP1-GFP* transgene. Heterozygotes were excluded to avoid confounding by dominance effects in this analysis. The points show the petal numbers of the individual F2 plants and the lines the fitted linear model. Note that the regression coefficients are different depending on the presence of *gAtAP1-GFP* and that these tend to be larger in its absence, in agreement with epistasis.

</Author response image 1 title/legend>

3) The authors propose that differences in AP1 expression between species could cause the phenotype. Using qPCR data, they show that the ChAP1 expression is lower than AtAP1 expression in plants carrying 2 or 4 copies, but does that affect the phenotype?

By quantifying *AP1* expression, transgene copy number, and petal number in our *gAtAP1-GFP* and *gChAP1-GFP* lines we were able to test whether *AP1* dose linearly affected petal number – it doesn’t. However, as we describe in Figure 6A and associated text, there are many reasons to expect a non-linear, rather than linear, relationship between *AP1* expression and petal number (see Felix and Barkoulas, 2015). These complex problems have recently started to be investigated in other systems. For example, direct measurement of single mRNA molecules has been used to quantify a nonlinear dose-response curve between gene activity and phenotype (Barkoulas et al., 2013), and could address the issues raised by the reviewer. However, methodologies for such quantitative analysis have not yet been reported in flowers. This interesting line of investigation could be the topic of a future, dedicated study and is well beyond the scope of this current paper.

The results in Figure 4C are difficult to interpret, due to the high level of variability in petal number for all constructs (even the ap1 mutant complemented with pAtAP1::AP1, contrary to the equivalent construct in Figure 3A where the distribution appears tightly around 4).

We maintain that the results we present in Figure 4C are in fact quite clear. We have chosen to show all the data in beeswarm plots throughout the manuscript. This allows the reader to interact directly with the data, and as we are studying a variable phenotype, a feature of our data is variability. It would look a lot cleaner if we summarized only the mean and standard error! However, we show the means and ANOVA results to support our conclusion that there is a significant effect of promoter sequence on petal number (*p* = 9.45e-08) but no effect of coding sequence (*p* = 0.103) and no interaction effect between the promoter and coding sequences (*p* = 0.258).

It is important to note that the constructs used in Figure 4C differ to those in Figure 3A. The *A. thaliana (gAtAP1-GFP*) and *C. hirsuta (gChAP1-GFP*) constructs used in Figure 3A carry the full genomic regions of *AP1*, including introns. In Figure 4C, we constructed chimeras between *AP1* promoter and cDNA sequences, which lack introns. Therefore, *pAtAP1::AtAP1*, used in Figure 4C, is not strictly equivalent to *gAtAP1-GFP*, used in Figure 3A, which might explain the difference between results.

But when looking at a double AtAP1 and ChAP1 marker in Arabidopsis, the authors also observe changes in expression in the area between sepals (which is not seen in Cardamine, based on the picture in Figure 4E and F and in Figure S5). Please discuss these differences. Is more AP1 needed in the second whorl or less AP1 between sepals in the first whorl to promote petal formation? An analysis of the pattern of expression of AtAP1 and ChAP1 in Arabidopsis similar to the one performed in Cardamine using the double reporter might help.

First, the difference in accumulation of ChAP1 vs AtAP1 in the region between sepals is found in both *A. thaliana* and *C. hirsuta* (Author response image 2, marked by blue dashed line). This difference is quantitative so it can’t be observed by comparing single marker lines, only by using a double marker line. In contrast, expression (AtAP1) vs absence (ChAP1) in the second whorl is a qualitative difference that can be compared easily between single marker lines (Figure 4E, F and Author response image 2, marked by white dashed line) and in double marker lines (Author response image 2, marked by white dashed line). To clarify this point, we have now included an image of a *C. hirsuta* flower expressing gAtAP1-GFP and gChAP1-RFP for comparison with the same genotype in *A. thaliana*, and present this co-localization data as a supplementary figure (Figure 4—figure supplement 7).

**Author response image 2. respfig2:** Author response image ChAP1 vs AtAP1 expression in *C. hirsuta* stage 4 flowers. Transgenic lines expressing gAtAP1-GFP and gChAP1-RFP (**a**), gAtAP1-GFP (**b**) and gChAP1-GFP (**c**). A quantitative difference between gAtAP1-GFP and gChAP1-RFP can be observed by the predominantly red signal, indicating higher gChAP1 expression, in the region between sepals (marked by blue dashed line in panel a). This quantitative difference can’t be observed by comparing single marker lines (marked by blue dashed lines in panels b and c). A qualitative difference between gAtAP1-GFP and gChAP1-RFP can be observed by the predominantly green signal, indicating higher gAtAP1 expression, in petal initiation sites on the floral meristem (marked by white dashed line in panel a). This qualitative difference can also be observed by comparing single marker lines (marked by white dashed lines in panels b and c).

Second, we have identified that sequences sufficient to alter petal number between *A. thaliana* and *C. hirsuta* lie in the upstream promoter region of *AP1* (Figure 4C). These are functional experiments, which we supplement with detailed observations of species-specific differences in *AP1* expression (Figure 4D-G, Figure 4—figure supplement 2). We acknowledge that these observations do not demonstrate causality. However, our expression analysis is quantitative and performed at high resolution with live imaging, therefore we believe it provides valuable data to supplement our functional experiments, although future work is required to fully address the reviewers query.

Please also compare these results with the ones in their previous article (Mc Kim et al., 2017), where they showed that changes in growth of the floral buds correlated with the variations in petal number observed in response to seasonal changes in temperature.

This is an interesting point and is the reason why we performed such extensive quantitative analysis of *gAtAP1-GFP* and *gChAP1-GFP* transgenic lines in *C. hirsuta* (Figure 4E-G; Figure 4—figure supplement 3-6). From our previous work (Mc Kim et al., 2017) we hypothesised that the size or shape of floral buds may be altered as a consequence of *gAtAP1* expression, resulting in four petals. However, we found that *AP1* expression increased in the second whorl of *gAtAP1* floral buds, without changing bud size or shape (Figure 4E-G; Figure 4—figure supplement 3-6). We have now added this comparison in the text: “This contrasts with the changes in growth and maturation of floral buds that are associated with the regulation of *C. hirsuta* petal number by seasonal changes in temperature (McKim et al., 2017).”

The analysis in Cardamine (Figure 4E to G) seems to suggest that ChAP1 is less expressed in the second whorl than AtAP1, which could cause the phenotype. Do Cardamine plants carrying more copies of CdAP1 also produce more petals and is the variability in petal number reduced? This would support the point raised by the authors.

Our analysis shows that expression in the second whorl is a qualitative difference that distinguishes AtAP1 from ChAP1 expression. The genomic constructs that we report in Figure 3A and Figure 4E, F express extra copies of ChAP1 and AtAP1 in *C. hirsuta.* The consequence of an extra copy of AtAP1 is the accumulation of AP1 in the second whorl (Figure 4E), an increase in petal number and loss of variability (Figure 3A). However, an extra copy of ChAP1 does not accumulate in the second whorl (Figure 4F), and does not reduce the variability in petal number (Figure 3A).